# Particulate Matter Concentrations over South Korea: Impact of Meteorology and Other Pollutants

Shaik Allabakash [1,2], Sanghun Lim [2,*], Kyu-Soo Chong [2] and Tomohito J. Yamada [3]

1    Centre de Météorologie Radar, Météo-France, 31100 Toulouse, France
2    Korea Institute of Civil Engineering and Building Technology, Goyang-si 10223, Korea
3    Faculty of Engineering, Hokkaido University, N13 W8, Kita-ku, Sapporo 060-8628, Japan
*    Correspondence: slim@kict.re.kr

**Abstract:** Air pollution is a serious challenge in South Korea and worldwide, and negatively impacts human health and mortality rates. To assess air quality and the spatiotemporal characteristics of atmospheric particulate matter (PM), PM concentrations were compared with meteorological conditions and the concentrations of other airborne pollutants over South Korea from 2015 to 2020, using different linear and non-linear models such as linear regression, generalized additive, and multivariable linear regression models. The results showed that meteorological conditions played a significant role in the formation, transportation, and deposition of air pollutants. $PM_{2.5}$ levels peaked in January, while $PM_{10}$ levels peaked in April. Both were at their lowest levels in July. Further, $PM_{2.5}$ was the highest during winter, followed by spring, autumn, and summer, whereas $PM_{10}$ was the highest in spring followed by winter, autumn, and summer. PM concentrations were negatively correlated with temperature, relative humidity, and precipitation. Wind speed had an inverse relationship with air quality; zonal and vertical wind components were positively and negatively correlated with PM, respectively. Furthermore, CO, black carbon, $SO_2$, and $SO_4$ had a positive relationship with PM. The impact of transboundary air pollution on PM concentration in South Korea was also elucidated using air mass trajectories.

**Keywords:** air pollution; generalized additive model; multivariable linear regression model; meteorology; particulate matter; transboundary air pollution; South Korea



## 1. Introduction

Poor air quality due to air pollution critically impacts human health and can lead to increased mortality rates as well as negative impacts on biodiversity, ecosystems, and regional climate [1]. Atmospheric particulate matter (PM) is a mixture of suspended liquid and solid particles that plays a significant role in air pollution. In recent decades, rapid industrialization and urbanization in East Asia have resulted in severe air pollution and frequent haze events in the region [2]. South Korea is a recently developed East Asian country located downwind of China, Mongolia, and Russia. Elevated PM concentrations occur in South Korea due to emissions from local traffic, industrial facilities, and the long-range transport of anthropogenic emissions, Asian dust, and wildfire ash [3]. PM concentrations vary depending on human activities, topography, and meteorological factors [4]. In particular, meteorological parameters play a significant role in the dispersion, transformation, and removal of air pollutants from the atmosphere [5]. Consequently, a regional meteorology analysis is required to fully describe the characteristics of PM [6].

Many studies have been conducted around the world to examine the physical and chemical properties of atmospheric PM, as well as the relationship between PM and meteorological parameters [6–11]. PM concentrations were negatively correlated with wind speed (WS), precipitation, and humidity, and positively correlated with pressure over China [12]. Furthermore, it is known that PM concentrations vary seasonally. Li et al. [13]

found high $PM_{2.5}$ and $PM_{10}$ concentrations in winter and autumn and low concentrations in spring and summer in Shenyang, Northeastern China. Caramagna et al. [14] observed that fine particle concentrations ($PM_{2.5}$ and $PM_{10}$) were higher during working (high traffic) days and lower during non-working days over Catania (Italy). Brusca et al. [15] described the plume's characteristics and the relationship between the plume and obstacles of various shapes using laboratory experiments.

Several researchers have investigated the variability of PM concentrations in South Korea. Kim et al. [3] observed that the inter-annual PM concentration was negatively correlated with WS variations over the Seoul metropolitan area. Furthermore, $PM_{10}$ levels peaked in the spring and were positively correlated with $SO_2$, $NO_2$, and CO [11]. The Korea–United States Air Quality (KORUS-AQ) field campaign was conducted in South Korea from May–June 2016. This field campaign investigated the factors controlling air quality over the South Korean region using observations from three aircraft, an extensive ground-based network, and three ships as well as an array of air quality forecast models [16,17]. The campaign showed that the $PM_{2.5}$ concentrations were higher over central Seoul than in coastal regions during transboundary transport days, due to gaseous urban emissions [18]. The transboundary pollution from China was mixed with the local pollution, and the heterogeneous chemical reactions produced high $PM_{2.5}$ concentrations [19]. Foreign and domestic contributions have different effects on PM concentration in South Korea. Kumar et al. [20] investigated the contributions of foreign pollution (particularly from China) to $PM_{2.5}$ levels in South Korea. They found that the Chinese anthropogenic contributions were greatest during the winter and spring seasons. The highest contribution was around 60% in January and February, and the lowest was in August. China accounted for 45% of annual anthropogenic emissions. They also stated that domestic contributions were lower than Chinese contributions both monthly and annually, and contributions from the rest of the world were minor. Xie and Liao [21] examined the effects of changes in anthropogenic emissions in China on $PM_{2.5}$ concentrations in South Korea and Japan. The reduction in anthropogenic emissions in China resulted in a 7.7% and 7.9% decrease in $PM_{2.5}$ concentrations in South Korea and Japan, respectively. Additionally, the reduction in anthropogenic emissions in China reduced the concentrations of sulfate, ammonium, BC, and OC in Korea and Japan. During the KORUS-AQ campaign, which was conducted from May 1 to June 12, 2016, Lee et al. [22] examined the impact of long-distance transboundary transport on the Korean peninsula. They concluded that anthropogenic emissions in East-Central China, particularly in the Shandong region, have a significant impact on aerosol pollution in the Korean region. Yim et al. [23] described the impact of local and transboundary air pollution on air quality in Japan and South Korea. According to the authors, local emissions in both Japan and Korea accounted for 30% of the pollution, while transboundary emissions, specifically emissions from China's industrial sectors, accounted for the remaining 70%. Transboundary air pollution increased during favorable weather conditions (westerly or northwesterly winds) in the spring and winter seasons. Choi et al. [24] explored the effects of local and transboundary air pollution on $PM_{2.5}$ concentrations in Korea during the months of May and June of 2016. During the extreme pollution period, the Chinese contribution was 68% and the local contribution was 26%. Chinese and domestic contributions were 25% and 57%, respectively, during the blocking period. Furthermore, they discussed the role of emission sources in pollution reduction: the most prominent were domestic anthropogenic $NH_3$ emissions, followed by anthropogenic $SO_2$ emissions from Shandong, domestic anthropogenic $NO_X$, anthropogenic $NH_3$ emissions from the Shandong region, domestic anthropogenic OC emissions, and domestic anthropogenic BC emissions. Park [25] observed a link between particulate matter concentration and mortality rate in Seoul, Korea. The author found that PM, $NO_2$, and $O_3$ concentrations were high during the winter and spring, resulting in a high probability of mortality. Deaths were unlikely in areas with low levels of PM, $NO_2$, and $O_3$. Pollutants have a significant impact on human health during extreme weather events (cold and heat waves). Han et al. [26] discovered high PM concentrations in the western and northwestern areas of the Seoul

metropolitan region during winter haze events. The pollution is primarily attributed to foreign pollution (particularly Chinese emissions), but local pollution also plays an important role under high pressure with weak vertical mixing. Using clustering analysis, Lee et al. [27] identified the source regions for high $PM_{10}$ concentrations in the Seoul region. High $PM_{10}$ concentrations in Seoul were caused by external sources such as the Gobi desert area and major cities and industrial districts in inland China, as well as internal sources such as the Yellow Sea and inland Korea. According to Lee et al. [28], meteorological conditions such as atmospheric pressure distribution, air current movement, and atmospheric stability change can all play important roles in the occurrence and persistence of high-concentration air pollution episodes in the Korea region. The majority of the pollution transported from the Eastern China region is transported under favorable westerly conditions.

In recent years, South Korea has experienced significant industrial and economic growth, resulting in frequent and severe haze events and increased mortality associated with pollution. Despite the government's efforts to control air pollution, poor air quality remains a serious issue. The KORUS-AQ field study examined the impact of meteorology on the relative influence of local and transboundary pollution and described the causes of poor air quality in South Korea. However, the KORUS-AQ study was conducted for only a short period of time during the spring, from 1 May to 10 June 2016. Thus, these studies were narrowly focused on a specific time period and were less concerned with the impact of various meteorological parameters on $PM_{2.5}$ and $PM_{10}$. Thus, the present study more comprehensively discusses the $PM_{2.5}$ and $PM_{10}$ characteristics across South Korea over a long period (2015–2020) and supports the findings of previous studies in this region. Furthermore, the monthly, seasonal, and annual distributions of PM were described, as well as the impact of various meteorological parameters, including the influence of local and foreign pollution at different time scales. We investigated the relationship between PM concentrations (both $PM_{2.5}$ and $PM_{10}$), other air pollutants (CO, black carbon (BC), $SO_2$, $SO_4$, and $O_3$), and various meteorological parameters (temperature (T), specific humidity (QLML), WS, pressure (P), precipitation, wind direction, heat flux, and ground heating) for different time scales. To achieve this, we used various techniques and tools such as the Spearman correlation, generalized additive model (GAM), and multivariable linear regression model (MLR). Further, we used Hysplit model outputs to define and interpret the impact of transboundary pollution on the Korean peninsula.

## 2. Materials and Methods

Since the 1960s, the East Asian region has experienced rapid industrialization, urbanization, and economic growth [29], resulting in increased air pollution in all East Asian countries, including South Korea. To measure and monitor air pollution, the Korean government established the Korean Ministry of Environment (KMOE) Network in 1973, which is controlled by national and local government bodies. The network stations monitor various types of pollution (air quality), including city air quality, road-side air quality, acid deposition, background density, suburban air quality, heavy metals, harmful materials, photochemical pollutants, the global atmosphere, PM concentrations, and air pollution concentrations [30].

Figure 1 depicts South Korea's 16 administrative areas, which include nine provinces and seven major cities. The KMOE conducts continuous automatic monitoring of PM concentrations in all 16 administrative areas using various monitoring equipment. We used KMOE data to investigate the relationship between PM concentration and meteorology in Korea. The KMOE network provides six different types of air pollutants, including $PM_{10}$, $PM_{2.5}$, $O_3$, CO, $SO_2$, and NO. The air pollution-related information and data are available at www.airkorea.or.kr (accessed on 8 July 2021). The network maintains data accuracy and quality and filters out abnormal data in four stages. Abnormal data are identified based on the conditions of the (1) measuring equipment (e.g., fault connection, malfunction, calibration) and if the data exceeds the normal range or rate of change, (2) if the data acquisition rate is less than 75%, (3) if there are large fluctuations in the

data, and (4) if the data are inconsistent; and then the abnormal data filters automatically using inbuilt algorithms [31]. The accuracy of the data has been verified and used by several researchers for various air quality studies [3,11]. We also compared KMOE data with MERRA re-analysis datasets and found good agreement (figure not shown).

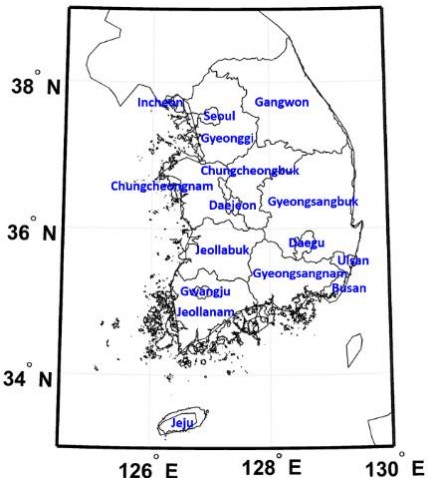

**Figure 1.** Map of South Korea showing the provinces and major cities.

In the present study, we used $PM_{2.5}$ and $PM_{10}$ measurements from 220 monitoring stations across all administrative areas for six years (2015–2020). Monthly $PM_{2.5}$ and $PM_{10}$ data were converted into seasonal, inter-annual, and annual time scales for the 16 administrative areas and for the entire country. We used various meteorological parameters (T, QLML, precipitation, WS, horizontal (zonal U) and vertical (W) wind vectors, ground heating, pressure, and latent and sensible heat fluxes); the surface concentrations of pollutants including CO, BC, $SO_4$, and $SO_2$; aerosol optical depth (AOD); and the mass mixing ratio of $O_3$ to explore their relationships with PM. Meteorological and pollutant parameters were collected from the state-of-the-art MERRA-2 reanalysis dataset for South Korea. Kim et al. [32] validated MERRA-2 data for the Korean Peninsula and found that it outperformed other reanalysis datasets. Several researchers have also used MERRA-2 data in different studies [33,34]. MERRA-2 data were developed and generated by the global modeling and assimilation office (GMAO) of the National Aeronautics and Space Administration (NASA) using the Goddard Earth observing system, version 5.12.4 data assimilation system [35]. MERRA-2 provides data at a $0.5° \times 0.625°$ (latitude $\times$ longitude) spatial resolution and various temporal resolutions (one hourly minimum time step). In this study, monthly data were used to evaluate seasonal, inter-annual, and annual values for meteorological and pollutant parameters. In addition, the data for each province and the entire country were averaged.

## 2.1. Generalized Additive Model (GAM)

Simple linear regression models are frequently used in statistical applications. However, linear models often fail to accurately fit non-parametric or non-linear parameters [36]. GAMs are more advantageous for non-parametric applications because they can describe and interpret non-linear parameters due to their additive effects. GAM supports a wide range of distributions and link functions, which reduces the error in predicting the dependent variable. GAMs are effective at handling complex non-linearity in air pollution research and have been widely used to detect the relationship between air pollutants and meteorological parameters [36–41]. GAMs are regression models that use smoothing splines for covariates rather than parametric or linear coefficients [36,42]. The additive model in the context of time series concentration can be described as [42].

$$\log(y_i) = \beta_0 + \sum_{j=1}^{n} s_j(x_{ij}) + \varepsilon_i$$

where $y_i$ is the ith air pollution concentration, $\beta_0$ is the overall mean of the response, $s_j(x_{ij})$ is the smooth function of the ith value of covariate j, n is the total number of covariates, $\varepsilon_i$ is the ith residual, and $var(\varepsilon) = \sigma^2$, which is assumed to be normally distributed. We used GAM to analyze the relationship between PM and meteorological parameters and other pollutants. The Akaike Information Criterion (AIC) and adjusted $R^2$ were used to select the degrees of freedom, air pollutants, and the appropriate meteorological parameters [43]. The smoothing parameters were selected based on the restricted maximum likelihood (REML) method. The proposed adjusted model in the present study is as follows:

$$\log(PM_k) = \beta_0 + s(T) + s(prec) + s(WS) + s(pressure) + s(BC) + s(CO) + s(SO_4) + s(OZ) + \varepsilon_i \tag{1}$$

where $PM_k$ represents either $PM_{2.5}$ or $PM_{10}$. The meteorological parameters, pollutants, and PM were selected based on the smallest AIC and the largest adjusted $R^2$, and cubic regression spline smoothing functions were used for meteorological factors, which were determined based on the REML standard All analyses were carried out in R using the "mgcv" package [44].

### 2.2. MLR Model

MLR model was used to delineate the relationships between multiple independent meteorological variables and a single dependent variable (pollutants). The MLR model was computed based on the ordinary least squares (OLS) method. The model is as follows:

$$Y_i = \beta_0 + \beta_1 x_{1i} + \beta_2 x_{2i} + \beta_3 x_{3i} + \beta_4 x_{4i} + \beta_5 x_{5i} + \varepsilon \tag{2}$$

where i represents the site locations in Korea; $\beta_0$, $\beta_1$, $\beta_2$, $\beta_3$, $\beta_4$, $\beta_5$ are the regression coefficients; y is the concentration of the independent pollutant variable; x is the meteorological variables ($x_1$, $x_2$, $x_3$, $x_4$, and $x_5$ are the QLML, P, WS, T, and precipitation, respectively); and $\varepsilon$ is the stochastic error associated with the regression.

### 2.3. HYSPLIT Model

The National Oceanic and Atmospheric Administration (NOAA) developed the hybrid single-particle Lagrangian Integrated Trajectory (HYSPLIT) to determine the transport paths of air masses. This model has been widely used to investigate the transport of air pollutants using forward- and back-trajectories. The model simulates air parcel trajectories, emission, difficult transport, deposition, dispersion, and chemical transformation. The NOAA's global data assimilation system (GDAS) $1° \times 1°$ grid resolution data were used to examine transboundary air pollution. For trajectory calculation, the model employs the NCEP/NCAR re-analysis meteorological dataset. Air mass trajectories define the transportation of aerosols from one location to other [45]. At specific weather conditions, the air mass carries a high number of aerosols and pollutants, which illustrates the heavy pollution or haze events. In the present study, we used 24 h back-trajectories during haze and non-haze events. The air mass backward trajectories were effective in determining the source of the pollution and its pathway to the receptor (long-range transportation) [45].

### 3. Results

#### 3.1. Monthly Distribution of PM and Other Pollutants

Figure 2a,b show the monthly mean variability of $PM_{2.5}$ and $PM_{10}$ over South Korea from 2015 to 2020, respectively. There is a clear V-shaped distribution for both $PM_{2.5}$ and $PM_{10}$ throughout each year. $PM_{2.5}$ concentrations were highest in December, January, and February, and lowest in July and August. In contrast, $PM_{10}$ concentrations were highest in March and April and lowest in July and August. $PM_{2.5}$ concentrations ranged from

10–40 µg m$^{-3}$ and were higher than the World Health Organization (WHO) standard of 25 µg m$^{-3}$. PM$_{10}$ concentrations ranged at 20–80 µg m$^{-3}$ and also exceeded the WHO recommended levels of 50 µg m$^{-3}$. High PM concentrations were caused by industrial emissions and dust transport from neighboring countries in conjunction with favorable meteorological conditions [3]. The peak PM$_{2.5}$ concentration (40 µg m$^{-3}$) was observed in 2019, due to heavy haze events. Generally, both PM$_{2.5}$ and PM$_{10}$ tended to decrease over the study period, with the lowest PM concentrations observed in 2020 attributed to the full and partial lockdowns in China and Korea to prevent the spread of the novel coronavirus (COVID-19). During this period, major industries were closed, and traffic significantly decreased, which had a considerable impact on emissions.

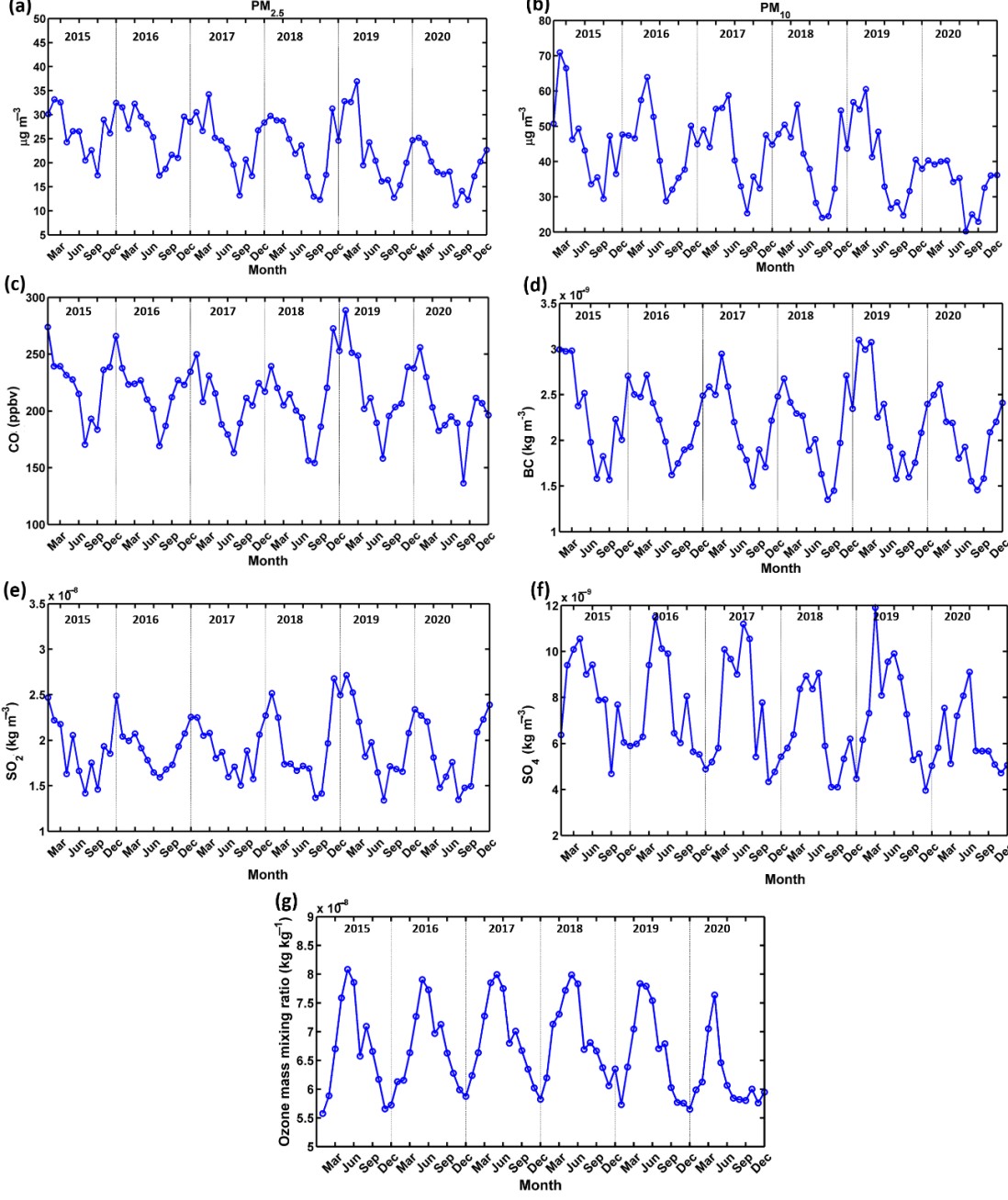

**Figure 2.** Monthly average concentrations of particulate matter (PM) and other pollutants over South Korea (2015–2020). (**a**) PM$_{2.5}$, (**b**) PM$_{10}$, (**c**) CO, (**d**) BC, (**e**) SO$_2$, (**f**) SO$_4$, and (**g**) ozone mass mixing ratio.

The monthly mean variability across South Korea for CO, BC, SO$_2$, SO$_4$, and O$_3$ are shown in Figure 2c–g, respectively. The concentrations of CO, BC, and SO$_2$ were high in

December and January and low in July and August, similar to the annual variations in $PM_{2.5}$. In contrast, $SO_4$ and $O_3$ levels were high in March and April and low in January and December. Peak values of CO, BC, $SO_2$, and $SO_4$ were detected in 2019, which were probable sources of the simultaneous high $PM_{2.5}$ concentrations. To examine the haze events and pollution that caused these peak values, we described the diurnal variations in $PM_{2.5}$, $PM_{10}$, and other pollutants in 2019. According to the Korea Meteorological Administration, South Korea experienced two long-lasting severe haze events in 2019: the first from January 11 to 15 and the second from February 20 to March 8 (especially 1–7 March 2019) [46]. Figure 3 shows the diurnal variabilities in PM and other pollutants for January–March 2019. During these haze events, high concentrations of $PM_{2.5}$ and $PM_{10}$ were observed, and peak values of 99.2 μg m$^{-3}$ and 144.3 μg m$^{-3}$, respectively, were detected on 4–5 March 2019. The concentrations of other pollutants (CO, BC, $SO_2$, and $SO_4$) were also higher during these events and peaked on 4–5 March 2019. However, there was no significant variation in $O_3$ levels. On 13–14 January 2019, $PM_{2.5}$ and $PM_{10}$ levels were 90 μg m$^{-3}$ and 120 μg m$^{-3}$, respectively. Furthermore, CO, BC, and $SO_2$ have high peak values, whereas $SO_4$ has a moderate peak value. The high concentrations of other pollutants support the haze events that result in high PM concentrations. Interestingly, the high and low trends of $PM_{2.5}$ and $PM_{10}$ were mostly consistent with CO, BC, $SO_2$, and $SO_4$, while $O_3$ differed. Figure S1 shows the AOD for these haze events. High AOD values were observed on 11–15 January 2019and 1–7 March 2019and were associated with PM and other pollutant concentrations. Furthermore, AOD was highest on 14 January 2019 (0.44) and 5 March 2019 (1.19) during the first and second events, which also matched the peaked values of PM. The synoptic meteorological conditions during the severe haze events (13–14 January and 4–5 March 2019) are shown in Figure S2. During these haze events, low temperature, low-to-moderate humidity, and northwesterly/westerly winds with a WS of 5–10 m s$^{-1}$ were observed in South Korea. These northwesterly/westerly winds possibly transported the pollutants from China to Korea. Figure S2 shows a high-pressure system located in South Korea and low-pressure systems located in Northern China, North Korea, and Japan. Because these low-pressure systems were extended and merged, wind flow was reduced, which resulted in the stagnation of pollutants over the region. Thus, the pollutants transported from China remained suspended for longer periods over South Korea [26,47] and were mixed with the local emissions [19]. Therefore, the high-pressure environment combined with weak vertical dispersion from winds led to PM accumulation.

Although local pollution prevailed in South Korea, we examined the 24 h backward trajectories of air masses arriving at four sites (Seoul, Jeollabuk, Ulsan, and Jeju) from 1 to 5 March 2019to assess the impact of transboundary air pollution during heavy haze events (See Figure 4). Despite the fact that Figure 4 only shows one eastern site, two western sites, and one individual (Jeju) site, we observed trajectory patterns at all of the sites. The western sites were significantly influenced by transboundary air pollution from Eastern and Northern China; the backward trajectories show that majority of airmasses were transported from Northern China (Harbin, Jilin, and Liaoning). Interestingly, during the severe haze days (4–5 March 2019), airmasses were transported from the Beijing, Hebei, and Tianjin regions; therefore, the severe haze events were primarily caused by emissions from Eastern China. Figure 5 illustrates the trajectory frequencies and percentage of airmass transport for the four sites from February 20 to March 10. Most of the airmasses arriving in Seoul and Jeollabuk originated in Northern and Eastern China, while those in Ulsan and Jeju arrived from all directions. Figure 6 depicts the trajectory frequencies for the January haze event. Seoul and Jeollabuk have a higher percentage of airmasses transported from Eastern and Northeastern China, whereas airmasses originated primarily in Northeastern China for Ulsan and Jeju. We examined the airmass patterns during non-haze days from January to March 2020.

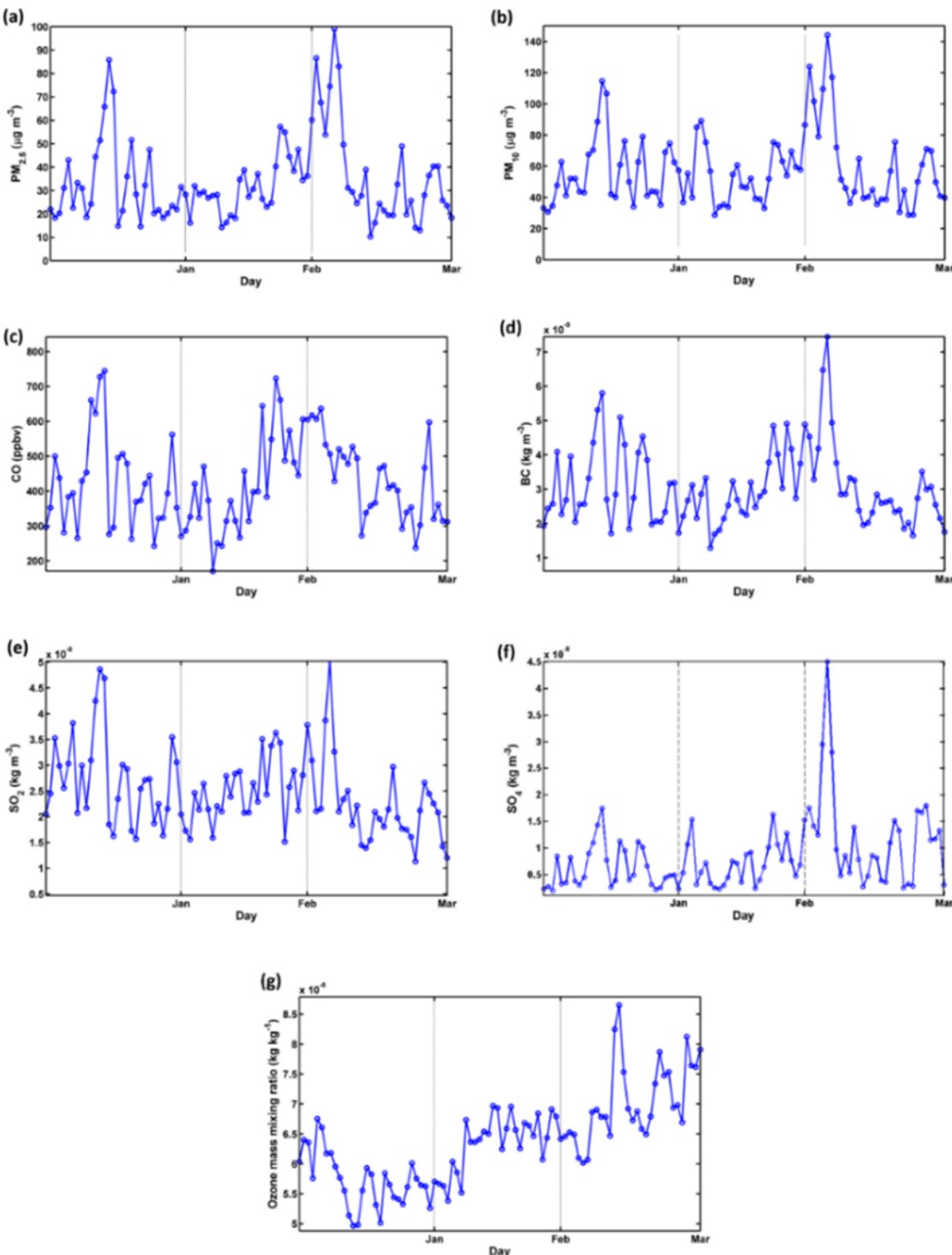

**Figure 3.** Diurnal distribution of particulate matter (PM) and other pollutants from January to March for 2019. (**a**) $PM_{2.5}$, (**b**) $PM_{10}$, (**c**) CO, (**d**) BC, (**e**) $SO_2$, (**f**) $SO_4$, and (**g**) ozone mass mixing ratio.

Figure S3 depicts the diurnal variation in PM concentrations during this time period. No severe haze events occurred in this period, and thus, peak PM concentrations did not occur as they did in January–March 2019 (Figure 3). PM concentrations were less than $60 \ \mu g \ m^{-3}$, and other pollutants were also in low concentrations (figure not shown). The corresponding trajectory frequencies in 2020 are shown in Figure S4 (20 February–10 March) and Figure S5 (1–15 January) for similar periods as those depicted in Figures 4 and 5 (2019). Figures S4 and S5 clearly show that the majority of air masses originated in Northeastern

China and the East China Sea, with no trajectories originating in Eastern China. According to the trajectory patterns for 2019 and 2020, transboundary air pollution in South Korea originated primarily in Northeast China and Mongolia. However, heavy transboundary air pollution originated mostly in Eastern China; Western Korea was also significantly influenced by this pollution.

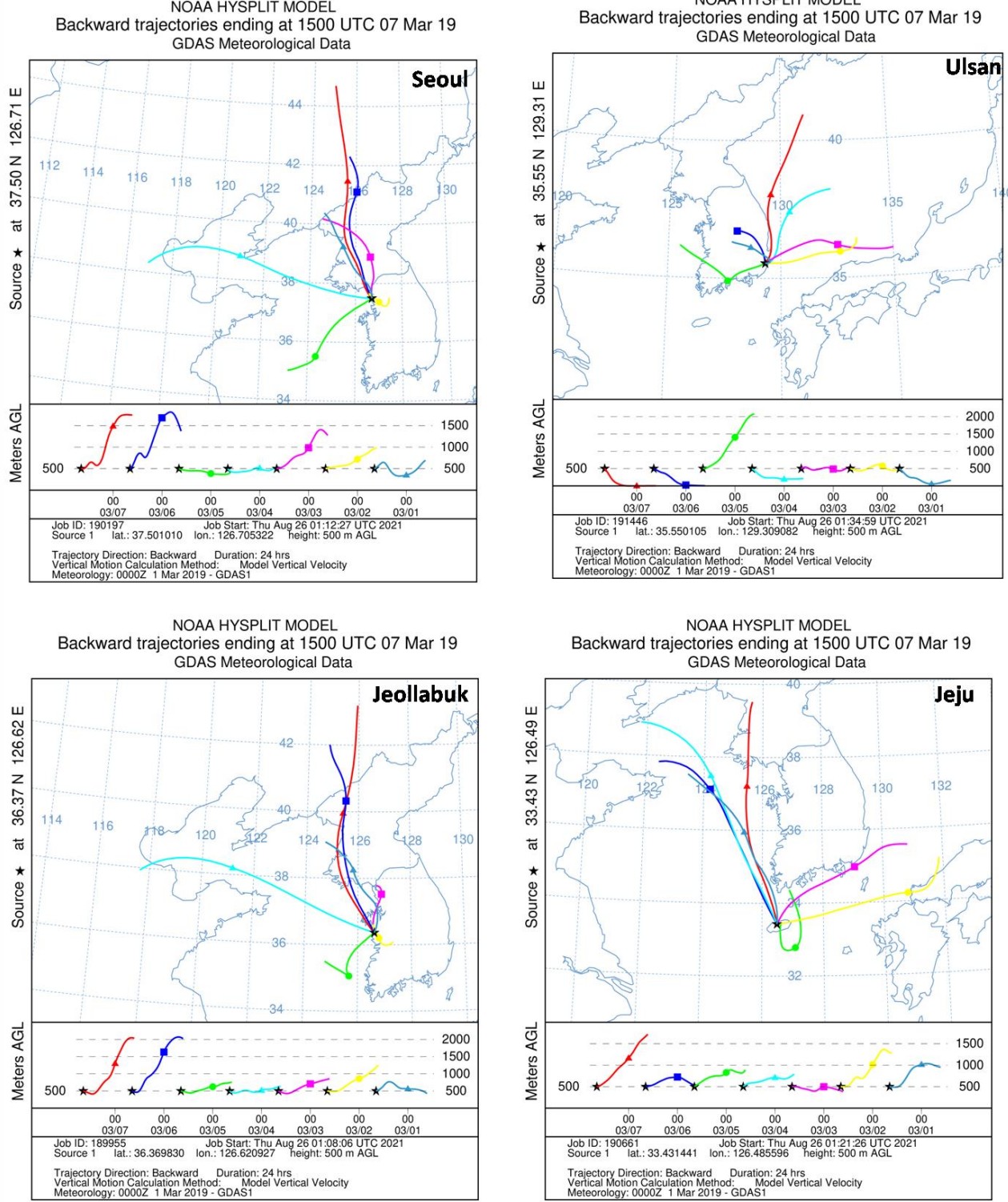

**Figure 4.** Backward trajectories of air masses derived from the Hysplit model for 1–5 March 2019 arriving at four sites in South Korea.

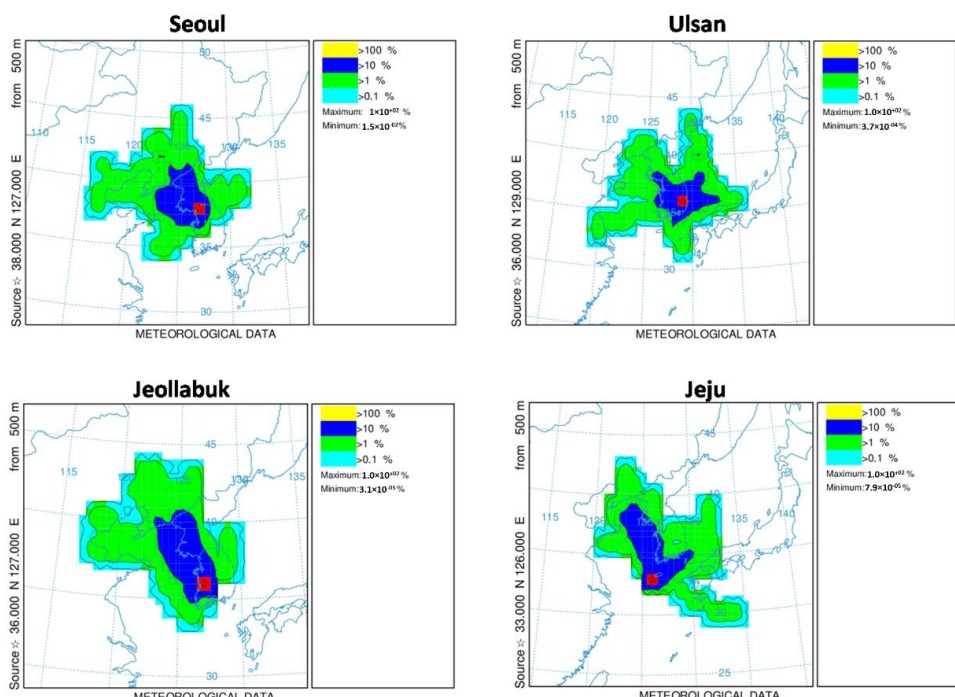

**Figure 5.** Trajectory frequencies (percentage of airmass transport) over the four sites from 20 February to 10 March 2019.

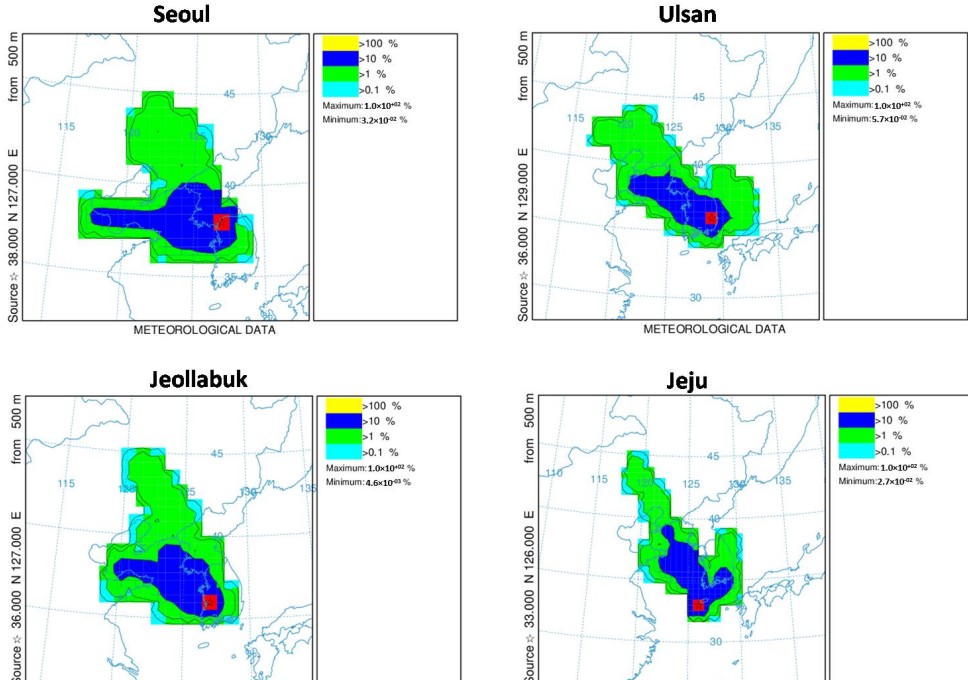

**Figure 6.** Trajectory frequencies (percentage of airmass transport) over the four sites during 1–15 January 2019.

### 3.2. Seasonal Variability

We estimated seasonal mean values for all air pollutants and PM. Figure 7 shows seasonal mean values of $PM_{2.5}$, $PM_{10}$, CO, BC, $SO_2$, $SO_4$, and $O_3$ for the entire country. $PM_{2.5}$ was the highest in winter followed by spring, autumn, and summer. CO, BC, and $SO_2$ were also high in winter and low in summer. $PM_{10}$ was the highest in spring, followed by winter, autumn, and summer. $SO_4$ and $O_3$ concentrations were also high in spring and interestingly, they were low during autumn. The high values of $PM_{2.5}$ in

winter were attributed to emissions from traffic, household heating and cooking, and industries, including coal-fired power plants, semiconductors, biopharmaceuticals, and cement [48]. The geography and climate of Korea, particularly wind direction, play an important role in PM concentration. Figure S6 shows the seasonal mean WS and direction over East Asia. Strong prevailing northwesterly winds blow over Korea from Northern China and Southern Mongolia, transporting anthropogenic emissions from these regions. In particular, the industrial regions of Harbin, Changchun, the Pearl Delta, Beijing–Tianjin, Shanghai, and Shenyang that are in Northeast and Eastern China are known to have high pollutant concentrations [2,6,13]. Thus, industrial emissions from have a significant impact on $PM_{2.5}$ concentrations in Korea. During the spring, the prevailing winds are westerly and northwesterly; consequently, dust is transported to Korea from China and Mongolia, increasing $PM_{10}$, $O_3$, and $SO_4$ concentrations. In summer, heavy precipitation reduces PM concentrations through wet removal and washing effects. Figure 2a,b show that the seasonal variation of PM concentrations decreased from 2015 to 2020.

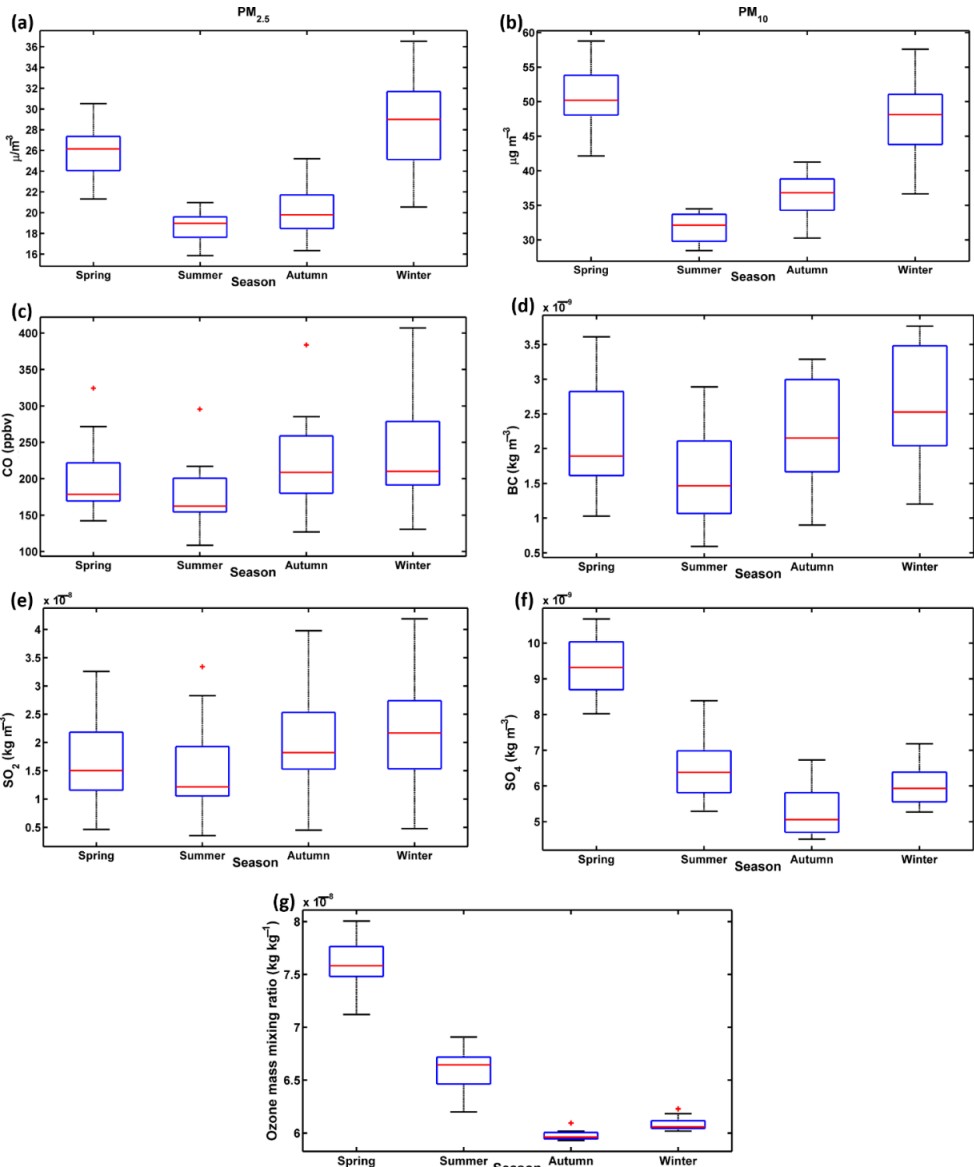

**Figure 7.** Seasonal mean values of particulate matter (PM) and other pollutants. The blue boxes represent the variations of the parameters, the lower and upper edges of the box are 25th and 75th percentiles, respectively, the red colored mark is the median, and the red plus symbols indicate outliers. (**a**) $PM_{2.5}$, (**b**) $PM_{10}$, (**c**) CO, (**d**) BC, (**e**) $SO_2$, (**f**) $SO_4$, and (**g**) ozone mass mixing ratio.

### 3.3. Annual Distribution of PM and Other Pollutants

Figure 8 shows the annual mean distributions of PM and other pollutants. The South Korean government has enacted several control measures and laws to improve air quality; as a result, pollutant concentrations have decreased from 2015 to 2020. However, pollution levels in South Korea remain higher than the WHO standards. Overall, the maximum $PM_{2.5}$ and $PM_{10}$ concentrations were 34 $\mu g\ m^{-3}$ and 54 $\mu g\ m^{-3}$, respectively. Furthermore, high values were detected in 2019 due to severe haze events (as described in the previous section). The mean values of $PM_{2.5}$ ($PM_{10}$) were 26.8 $\mu g\ m^{-3}$ (46.5 $\mu g\ m^{-3}$), 25.9 $\mu g\ m^{-3}$ (44.7 $\mu g\ m^{-3}$), 24.1 $\mu g\ m^{-3}$ (43.4 $\mu g\ m^{-3}$), 22.7 $\mu g\ m^{-3}$ (40.7 $\mu g\ m^{-3}$), 22.6 $\mu g\ m^{-3}$ (40.4 $\mu g\ m^{-3}$), and 18.4 $\mu g\ m^{-3}$ (33.5 $\mu g\ m^{-3}$) for 2015, 2016, 2017, 2018, 2019, and 2020, respectively. CO, BC, and $SO_4$ also tended to decrease from 2015 to 2020; however, $SO_2$ remained mostly consistent, and $O_3$ increased from 2015 to 2018 and then decreased.

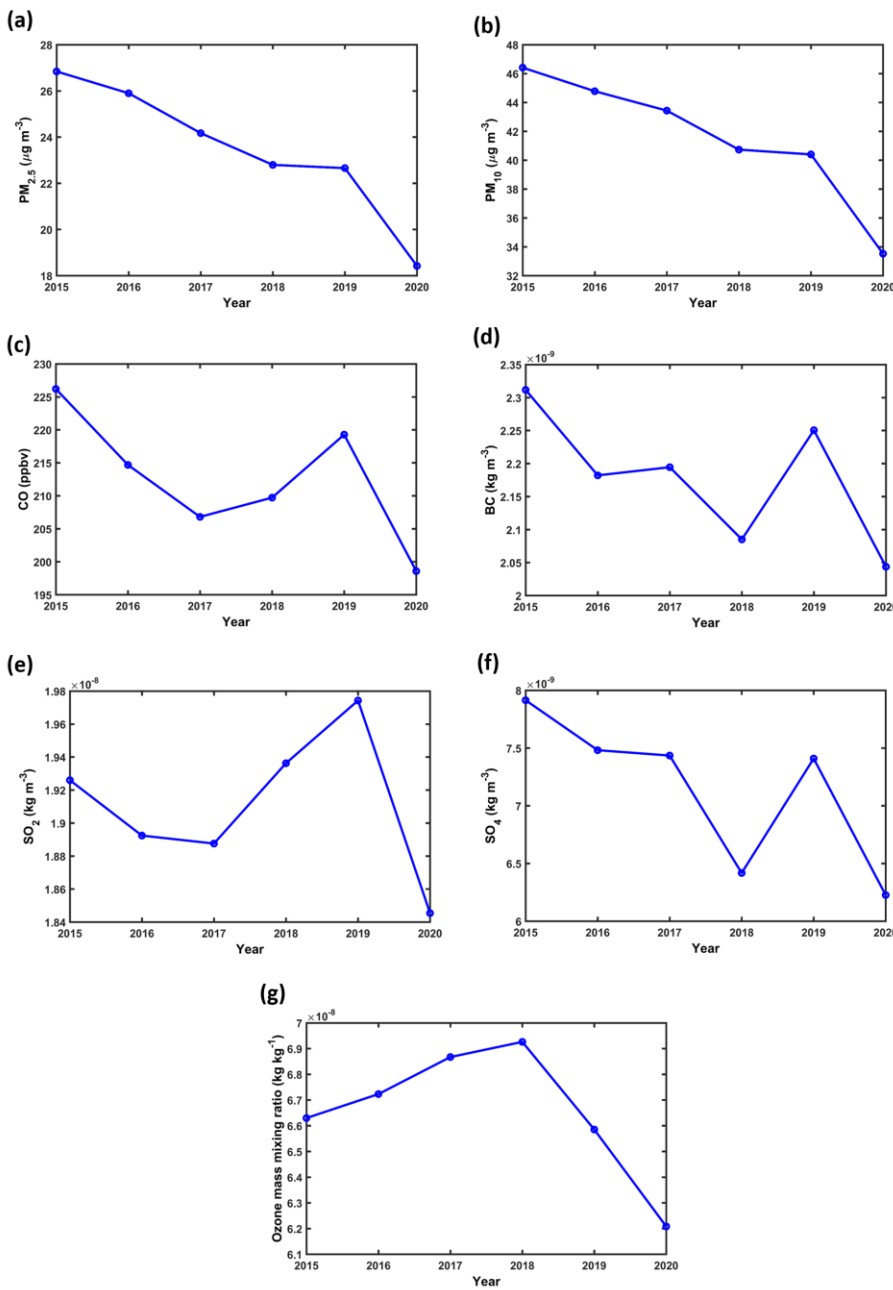

**Figure 8.** Mean annual distribution of particulate matter (PM) and other pollutants (2015–2020). (**a**) $PM_{2.5}$, (**b**) $PM_{10}$, (**c**) CO, (**d**) BC, (**e**) $SO_2$, (**f**) $SO_4$, and (**g**) ozone mass mixing ratio.

### 3.4. Spatial Distribution of PM and Other Pollutants

Monthly variations in $PM_{2.5}$ and $PM_{10}$ concentrations by administrative region are shown in Figure 9. All locations show an overall downward trend from 2015–2020. $PM_{2.5}$ concentration was high during winter and low during summer. In 2019, Chungcheong-buk recorded the highest $PM_{2.5}$ concentration (48.2 µg m$^{-3}$), followed by Gyeonggi (47.8 µg m$^{-3}$), Jeollabuk (45.4 µg m$^{-3}$), and Seoul (45.3 µg m$^{-3}$). These were attributed to the severe dust storm that occurred during the year. Seoul is the capital city and the largest metropolitan area in South Korea; hence, traffic volume and population density are high in the city and the surrounding Gyeonggi province. Seoul is commonly grouped with Gyeonggi and Incheon; this region is referred to as greater Seoul.

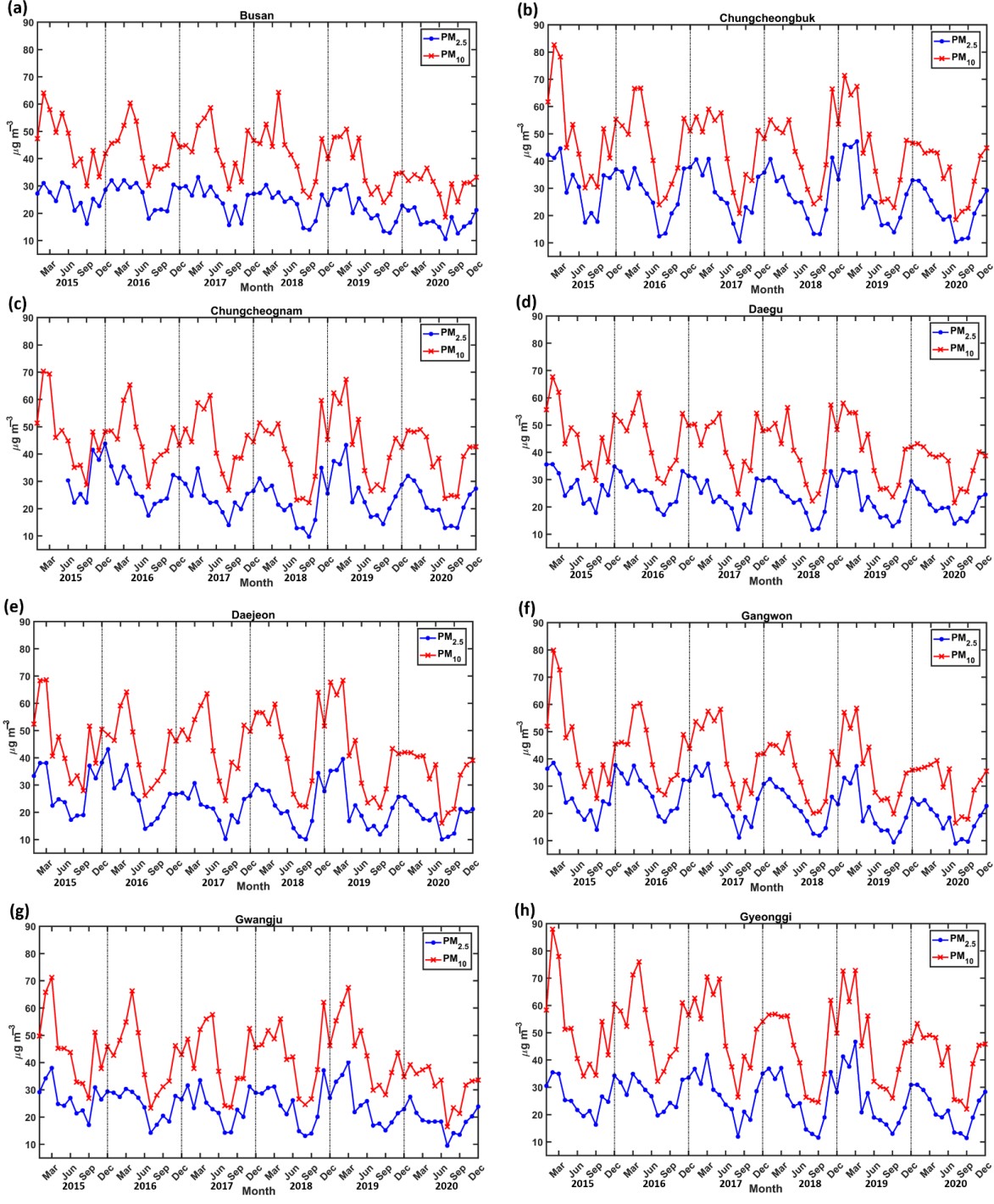

**Figure 9.** *Cont.*

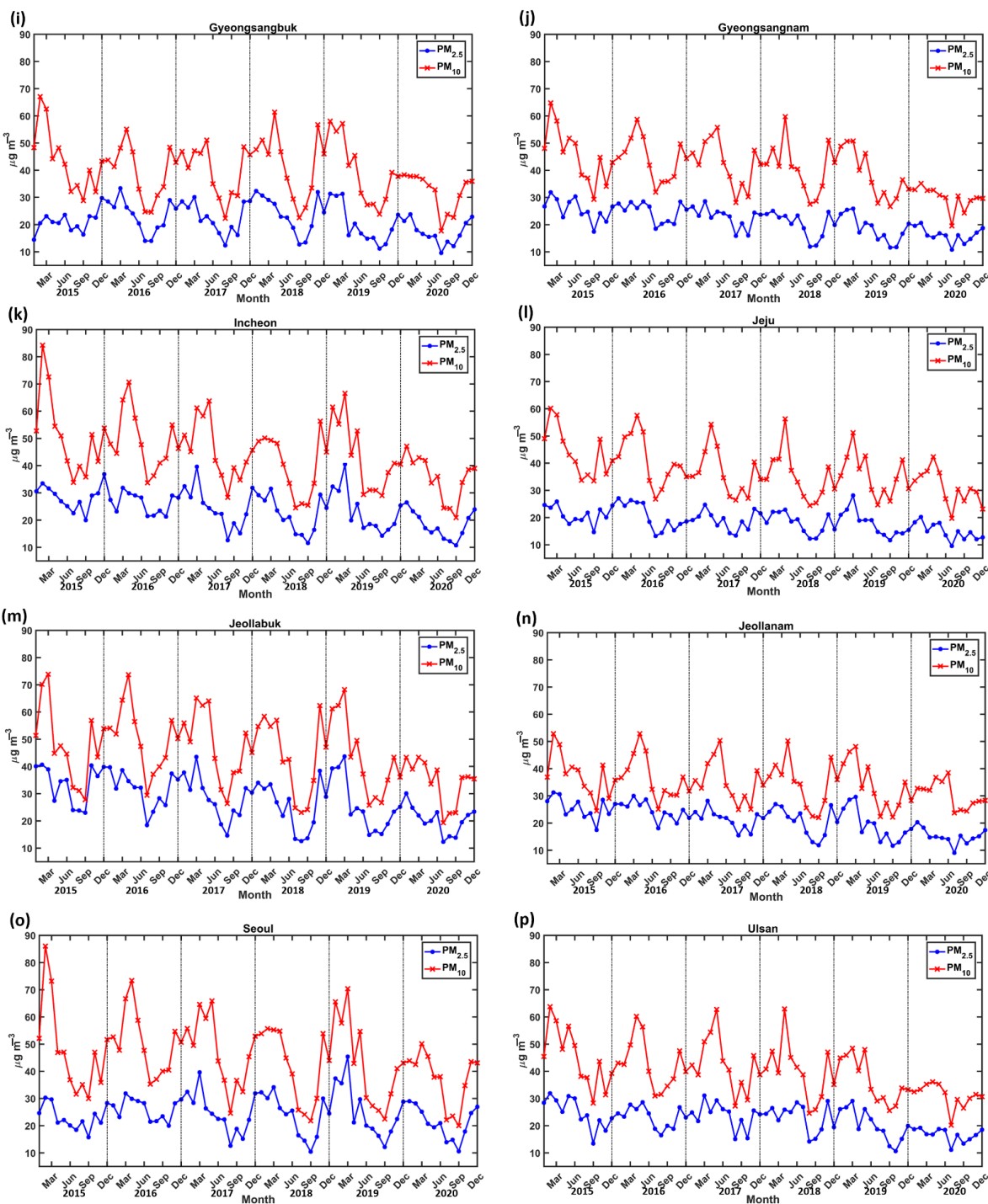

**Figure 9.** Monthly variations in PM$_{2.5}$ and PM$_{10}$ for the 16 administrative areas of South Korea from 2015 to 2020. (**a**) Busan, (**b**) Chungcheongbuk, (**c**) Chungcheognam, (**d**) Daegu, (**e**) Daejeon, (**f**) Gangwon, (**g**) Gwangju, (**h**) Gyeonggi, (**i**) Gyeongsanbuk, (**j**) Gyeongsangnam, (**k**) Incheon, (**l**) Jeju, (**m**) Jeollabu, (**n**) Jeollanam, (**o**) Seoul, and (**p**) Ulsan.

This highly urbanized area is characterized by high emissions from heavy traffic, industrial facilities, power generation, and household heating and cooking. Chungcheongbuk is located in the center of South Korea and on the southern side of the greater Seoul region. During the winter and spring, the prevailing winds from greater Seoul carry dust aerosols toward Chungcheongbuk. Furthermore, a diagonal mountain range to the southeast of Chungcheongbuk and Jeollabuk prevents further transport of these aerosols, resulting in

stagnation and accumulation of pollution. Bioindustrial and semiconductor industries are in Chungcheongbuk, whereas light industries are in Jeollabuk. Consequently, $PM_{2.5}$ concentrations were high in these regions. The record high $PM_{10}$ concentration was in Gyeonggi (89.6 µg m$^{-3}$), followed by Seoul (88.7 µg m$^{-3}$) and Incheon (88.4 µg m$^{-3}$) in 2015. $PM_{10}$ concentrations were high in the greater Seoul area compared to Jeollabuk and Chungcheongbuk. During the spring, all stations recorded high $PM_{10}$ concentrations. Interestingly, $PM_{10}$ concentrations were also high in Ulsan. Because this region has the largest shipyard and numerous oil refineries, dust emissions were high. Interestingly, the southernmost regions of Korea (Jeollanam and Jeju) had the lowest $PM_{2.5}$ (60 µg m$^{-3}$ and $PM_{10}$ (30 µg m$^{-3}$) concentrations. Jeollanam lacks major industries and was less influenced by the other regions (due to its location), and Jeju island is separated from the other Korean regions and thus less affected by pollution.

### 3.5. Relationships between PM, Pollutant Concentrations, and Meteorological Parameters

Meteorological conditions are the primary drivers of air pollution transport and dispersion [4,6,9,10,49]. Korea is surrounded by water on three sides and 70% of the land is mountainous. Owing to its complex topography, Korea has distinctive atmospheric conditions. The mountain winds, waves, and sea breezes significantly influence the weather. Figure 10 shows the seasonal distribution of the meteorological variables in South Korea that impact air pollution. Table 1 shows Spearman's rank correlation coefficients between the seasonal PM concentrations and pollutants and meteorological factors. The temperature was negatively correlated with both $PM_{10}$ and $PM_{2.5}$ in all seasons, and the relationship was significant during summer (−0.62 for $PM_{2.5}$ and −0.64 for $PM_{10}$). High temperatures favor intense mixing and turbulence, leading to vertical dispersion and improved air quality [33,50]. In winter, vertical dispersion was low (Figure 10h) due to more stable conditions. QLML and precipitation also had a negative correlation with both $PM_{10}$ and $PM_{2.5}$. The high QLML during summer promotes particle coalescence and leads to wet/dry deposition that reduces airborne PM [51]. Furthermore, high temperatures and QLML are conducive to strong convection, which causes heavy precipitation during summer. Heavy precipitation reduces $PM_{10}$ concentrations because raindrops entrain particles and remove aerosols from the atmosphere. Further, wind flow affects/reduces the $PM_{2.5}$ concentrations. Thus, the rainy weather can reduce PM concentrations. Therefore, PM has a stronger inverse relationship with QLML (−0.66 for $PM_{2.5}$ and −0.69 for $PM_{10}$) and precipitation (−0.49 for $PM_{2.5}$ and −0.47 for $PM_{10}$) during summer. Conversely, QLML produced weak negative correlations with PM in winter (−0.33 for $PM_{2.5}$ and −0.44 for $PM_{10}$), and precipitation had weak negative correlations with PM during spring (−0.15 for $PM_{2.5}$ and −0.03 for $PM_{10}$). Temperature, QLML, and precipitation showed significant negative correlations with $PM_{10}$ during autumn (−0.71, −0.78, and −0.61, respectively). There were some interesting correlations between WS and PM concentrations; in the summer and spring, WS was negatively correlated with $PM_{2.5}$ and $PM_{10}$, but slightly positively correlated in the autumn and winter. Although WS was generally low during spring and summer, the W wind vector was high, which facilitated vertical pollutant dispersion. In contrast, the U wind vector was low during the summer and spring, implying that the influence of horizontal dispersion was minimal. Allabakash and Lim [33] attributed higher planetary boundary layer heights during summer and spring to upward forcing. Conversely, W was low during winter and autumn, while U was high. Therefore, horizontal PM dispersion predominated during winter and autumn [33]. Table 1 shows that U was positively correlated with PM concentration, whereas W was mostly negatively correlated. The pressure was positively correlated with both $PM_{2.5}$ and $PM_{10}$ in all seasons except winter. Low pressure and vertical winds during spring and summer transport PM upward. Conversely, high pressure and low vertical winds tend to accumulate PM during winter [46,50]. Interestingly, the relationship between PM and ground heating was positive in the spring, summer, and winter, but negative in the autumn. We attribute this to soil moisture content; the soil is generally dry in the spring and wet in autumn. A similar relationship was observed for

latent and sensible heat fluxes. Finally, PM concentrations were positively correlated with BC, CO, SO$_2$, and SO$_4$ during all seasons. O$_3$ had weak correlations with PM in spring, autumn, and winter, but a strong positive relationship during the summer.

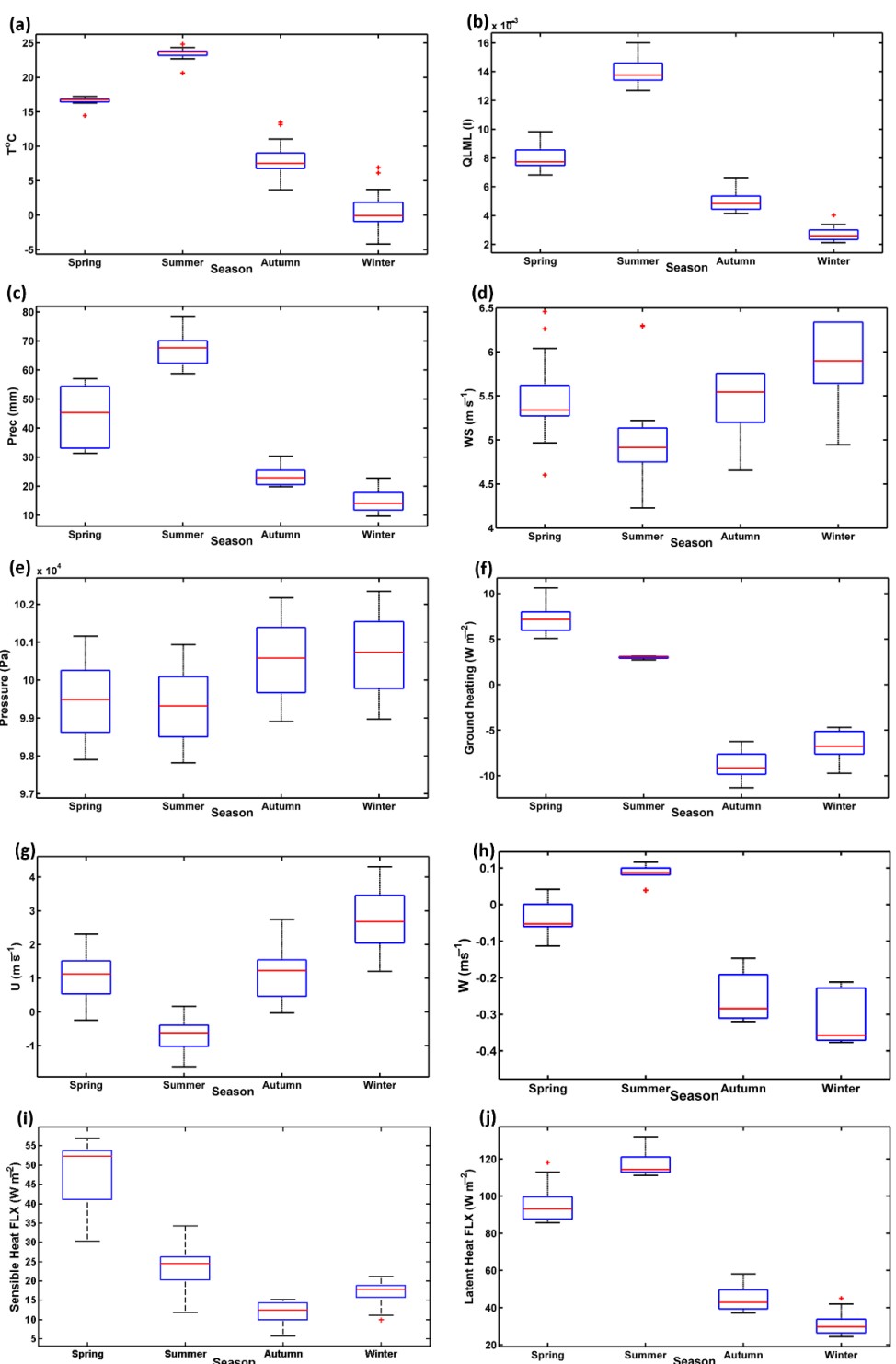

**Figure 10.** Seasonal variations of meteorological variables impacting air pollution in South Korea (2015–2020). The blue boxes represent the variations of the parameters, the lower and upper edges of the box are 25th and 75th percentiles, respectively, the red colored mark is the median, and the red plus symbols indicate outliers. (**a**) temperature, (**b**) specific humidity, (**c**) precipitation, (**d**) wind speed, (**e**) pressure, (**f**) ground heating, (**g**) horizontal (zonal) wind vector, (**h**) vertical wind vector, (**i**) sensible heat flux, and (**j**) latent heat flux.

**Table 1.** Spearman's rank correlation coefficients between the $PM_{10}$ and $PM_{2.5}$ concentrations and the meteorological factors and pollutants by season. Here, PM = particulate matter concentrations, T = temperature, QLML = specific humidity, Prec = precipitation, WS = wind speed, U = horizontal wind vector, W = vertical wind vector, P = pressure, GHT = ground heat.

| Season | PM | T | QLML | Prec | WS | U | W | P | GHT | BC | CO | $SO_2$ | $SO_4$ | $O_3$ |
|---|---|---|---|---|---|---|---|---|---|---|---|---|---|---|
| Spring | $PM_{2.5}$ | −0.42 | −0.51 | −0.15 | −0.18 | 0.11 | −0.39 | 0.43 | 0.11 | 0.77 | 0.61 | 0.64 | 0.64 | −0.24 |
| | $PM_{10}$ | −0.15 | −0.22 | −0.03 | −0.27 | 0.04 | −0.17 | 0.24 | 0.33 | 0.64 | 0.58 | 0.73 | 0.73 | 0.05 |
| Summer | $PM_{2.5}$ | −0.62 | −0.66 | −0.49 | −0.61 | 0.41 | −0.13 | 0.12 | 0.52 | 0.83 | 0.54 | 0.49 | 0.49 | 0.70 |
| | $PM_{10}$ | −0.64 | −0.69 | −0.47 | −0.57 | 0.47 | −0.13 | 0.06 | 0.49 | 0.85 | 0.59 | 0.56 | 0.56 | 0.66 |
| Autumn | $PM_{2.5}$ | −0.55 | −0.63 | −0.44 | −0.02 | 0.42 | −0.04 | 0.58 | −0.52 | 0.78 | 0.79 | 0.60 | 0.60 | −0.05 |
| | $PM_{10}$ | −0.71 | −0.78 | −0.61 | 0.12 | 0.56 | −0.22 | 0.72 | −0.70 | 0.88 | 0.87 | 0.76 | 0.76 | −0.25 |
| Winter | $PM_{2.5}$ | −0.19 | −0.33 | −0.27 | 0.01 | 0.05 | −0.01 | −0.26 | 0.30 | 0.75 | 0.52 | 0.19 | 0.19 | −0.10 |
| | $PM_{10}$ | −0.31 | −0.43 | −0.29 | 0.13 | 0.09 | 0.02 | −0.36 | 0.43 | 0.71 | 0.48 | 0.19 | 0.19 | 0.04 |

We used the annual distributions of each parameter to determine the dominant meteorological factor influencing PM concentration. The annual mean values of the pollutants and meteorological parameters for each year are shown in Figure 8 and Figure S7, respectively. $PM_{2.5}$ and $PM_{10}$ concentrations decreased from 2015 to 2020, but they were higher in 2019 than in 2018 and 2020. These figures show an upward trend for all parameters with a negative relationship with PM and vice versa. However, precipitation (Figure S7c) generally shows an upward trend, except for the very dry year in 2017, which does not agree with the trends in PM concentrations. Additionally, WS (Figure S7d) shows an upward trend, indicating a negative relationship with PM. Furthermore, WS was low in 2019, which suggests an inverse relationship with PM concentration. Based on these findings, we infer that WS was the dominant meteorological factor influencing PM concentration, in line with the findings of Kim et al. [3].

*3.6. Generalized Additive Model (GAM) Analysis*

GAM was used to describe the dependencies and non-linear relationships between PM and other pollutants and meteorological factors over South Korea during 2015–2020. Figure 11 shows the relationships for $PM_{2.5}$. Straight lines represent linear relationships, while curved lines represent non-linear relationships. $PM_{2.5}$ shows a strong negative dependency on temperature and precipitation. $PM_{2.5}$ has a negative relationship with low WS ($<6$ m s$^{-1}$) but no clear relationship with high WS ($>6$ m s$^{-1}$). Furthermore, $PM_{2.5}$ has a strong positive relationship with pressure, BC, and $O_3$, and a significant non-linear relationship with CO and $SO_4$. For low CO ($<200$ ppbv), the relationship was stable, whereas high CO ($>200$ ppbv) values were positively correlated. Interestingly, $PM_{2.5}$ was positively correlated with low $SO_4$ values and weakly negatively correlated with high $SO_4$ values. Figure 12 shows the GAM analysis for $PM_{10}$ dependencies, which are broadly similar to $PM_{2.5}$ dependencies. Precipitation and temperature show strong and weak negative relationships with $PM_{10}$, respectively, while pressure, BC, and $O_3$ show strong positive relationships. $PM_{10}$ has a weak relationship with WS and CO. $SO_4$ shows a non-linear relationship with $PM_{10}$. Overall, the relationships between PM and meteorological factors and other pollutant were consistent with the Spearman correlations shown in Table 1.

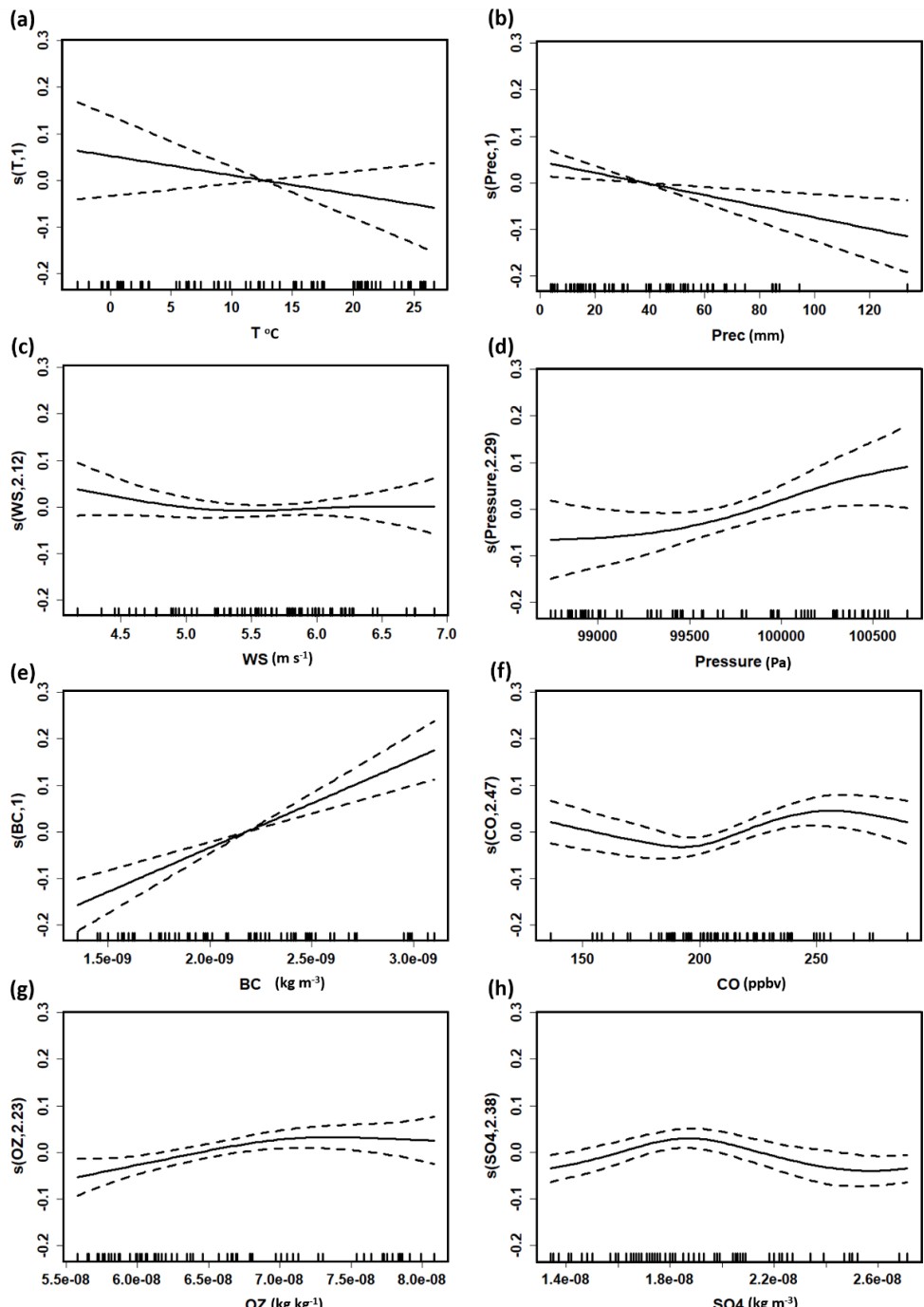

**Figure 11.** Generalized additive model (GAM) dependency analysis for $PM_{2.5}$ concentration. Response curves of $PM_{2.5}$ concentration to changes in (**a**) air temperature, (**b**) precipitation, (**c**) wind speed, (**d**) air pressure, (**e**) BC, (**f**) CO, (**g**) ozone mass mixing ratio, and (**h**) $SO_4$. The y-axis represents the smoothing function values (example s(T) is the trend in $PM_{2.5}$ concentration when air temperature changes, and the number (1) is the degree of freedom for the trend. The *x*-axis represents the measured values of the influencing factor, the solid line indicates the trend in $PM_{2.5}$ concentration with the change in influencing factors, and the dashed lines surrounding the solid line indicate the 95% confidence intervals.

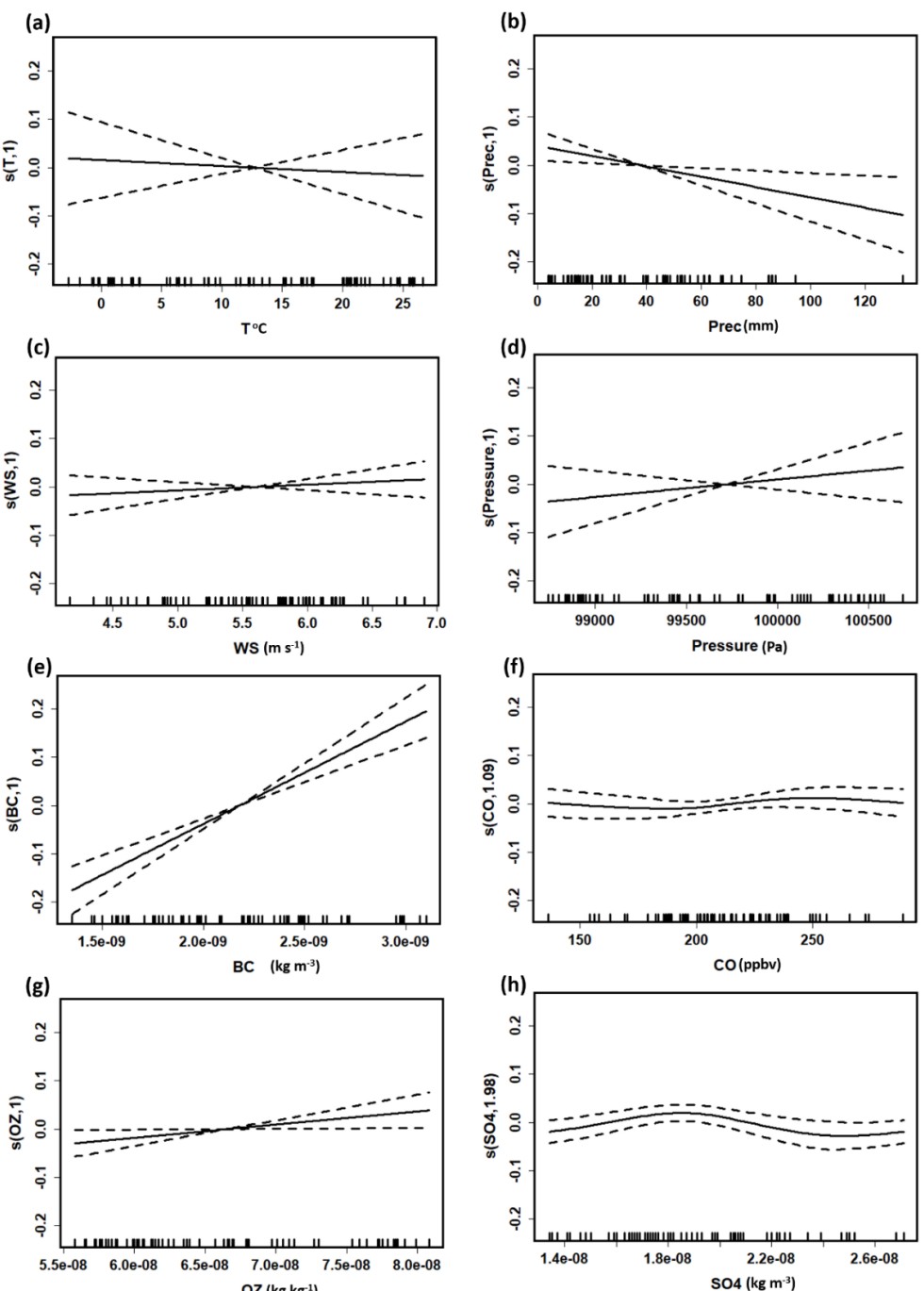

**Figure 12.** Generalized additive model (GAM) dependency analysis for PM$_{10}$ concentration. Response curves of PM$_{10}$ concentration to changes in (**a**) air temperature, (**b**) precipitation, (**c**) wind speed, (**d**) air pressure, (**e**) BC, (**f**) CO, (**g**) ozone mass mixing ratio, and (**h**) SO$_4$. The y-axis represents the smoothing function values (example s(T) is the trend in PM$_{10}$ concentration when air temperature changes, and the number (1) is the degree of freedom for the trend. The *x*-axis represents the measured values of the influencing factor, the solid line indicates the trend in PM$_{10}$ concentration with the change in influencing factors, and the dashed lines surrounding the solid line indicate the 95% confidence intervals.

*3.7. MLR Model Analysis*

Table 1 shows the linear correlations between PM, other pollutants, and meteorological factors during different seasons throughout South Korea. While the GAM model was used to describe non-linear relationships between these factors, the MLR model was used to

further examine the influence of meteorological factors on pollutants at 16 sites during different seasons. The independent variables for the MLR model were selected based on their low multi-collinearity and high $R^2$, F-test, and T-test values. Thus, QLML, P, WS, T, and precipitation were selected. The significance of the MLR model for $PM_{2.5}$ passed the F-test at 15 sites, with an average $R^2$ of 0.76 and at least four independent variables that passed the *t*-test. The F-test for $PM_{10}$ was also passed by 15 sites, with an average $R^2$ of 0.78 and at least four independent variables that passed the *t*-test. For concentrations of other pollutants (CO, BC, $SO_2$, $SO_4$, and $O_3$), the MLR model passed the F-test at 13 sites with an average $R^2$ of 0.74 and at least four independent variables that passed the *t*-test. The findings revealed that meteorological factors had distinct impacts in different parts of South Korea during different seasons.

Figure 13 depicts the effect of meteorological variables on $PM_{2.5}$ concentrations at all 16 sites during different seasons. During spring, WS and precipitation were the dominant factors that affected the $PM_{2.5}$ concentrations at all the sites except the eastern-most sites. Pressure also played a key role in $PM_{2.5}$ concentration changes over the central and eastern regions. In summer, temperature and precipitation were the dominant factors at most sites, and pressure was a significant factor at Ulsan and Busan. The WS influenced $PM_{2.5}$ levels in the central and southern regions during autumn (except Seoul and surrounding regions). WS was the main contributor to $PM_{2.5}$ levels in the northern and central regions (Gyeonggi and surrounding areas) during the winter. Temperature also had a strong influence on $PM_{2.5}$ in the southeast. Temperature and QLML influenced $PM_{2.5}$ at Jeju in the spring and winter, while pressure dominated in the summer.

The impacts of the meteorological variables on $PM_{10}$ concentrations are shown in Figure 14. During spring, WS, T, precipitation, and pressure were the dominant factors that influenced the $PM_{10}$ concentrations at most sites. During the summer, T and precipitation were the dominant factors in most sites, QLML and WS were dominant in the central sites, and P was dominant in the southeastern sites. $PM_{10}$ in the central and eastern regions was primarily influenced by WS; however, P was also influencing $PM_{10}$ over the eastern sites during autumn. In winter, WS was the dominant factor influencing $PM_{10}$ in Seoul, Jeollabuk, Gyengsonbuk, Daejon, Ulsan, Busan, and Daegu during the winter. T and P were the dominant factors that influenced $PM_{10}$ concentrations in Jeju during spring, summer, and winter, while precipitation, WS, and QLML influenced $PM_{10}$ concentrations in autumn. Figure S8 shows the impacts of the meteorological factors on CO. WS, T, P, and precipitation were the dominant factors impacting CO concentrations during all seasons at Seoul. Further, pressure significantly affected CO at all sites during all seasons. Over southeastern sites, T, P, and WS were dominant in the spring, and T and P were dominant in the winter, and these factors influence CO concentrations. In Jeju, CO concentration was influenced by T and P during spring and winter. However, WS was dominant in summer, and T and QLML were dominant during autumn. Figure S9 shows that, BC concentrations were significantly influenced by all the meteorological factors in the western part of Korea, but their influence was less in the east during all seasons. In Jeju, T was the dominant factor for BC concentration during all seasons. Furthermore, P and precipitation impacted the BC concentration during summer, while QLML impacted it during autumn and winter. Figure S10 shows the impacts of the meteorological factors on $SO_4$; they were mostly similar to those on BC. Figure S11 shows that the $SO_2$ concentrations over northwestern regions (especially the Seoul metropolitan region) were influenced by all meteorological factors, especially WS and precipitation, during all seasons. In Jeju, $SO_2$ was mostly influenced by T. $SO_2$ at the central sites was less affected by meteorological parameters during the summer. Figure S12 shows that the $O_3$ concentrations during spring were influenced by all the meteorological factors (mainly T, precipitation, and WS) in the western and southern regions. During the summer, all meteorological factors influenced $O_3$ concentrations in the northwest (Seoul and surrounding areas). Meteorological factors influenced $O_3$ concentrations at all sites during the autumn and winter seasons. $O_3$ concentrations in Jeju were influenced by T, P, and QLML during all seasons.

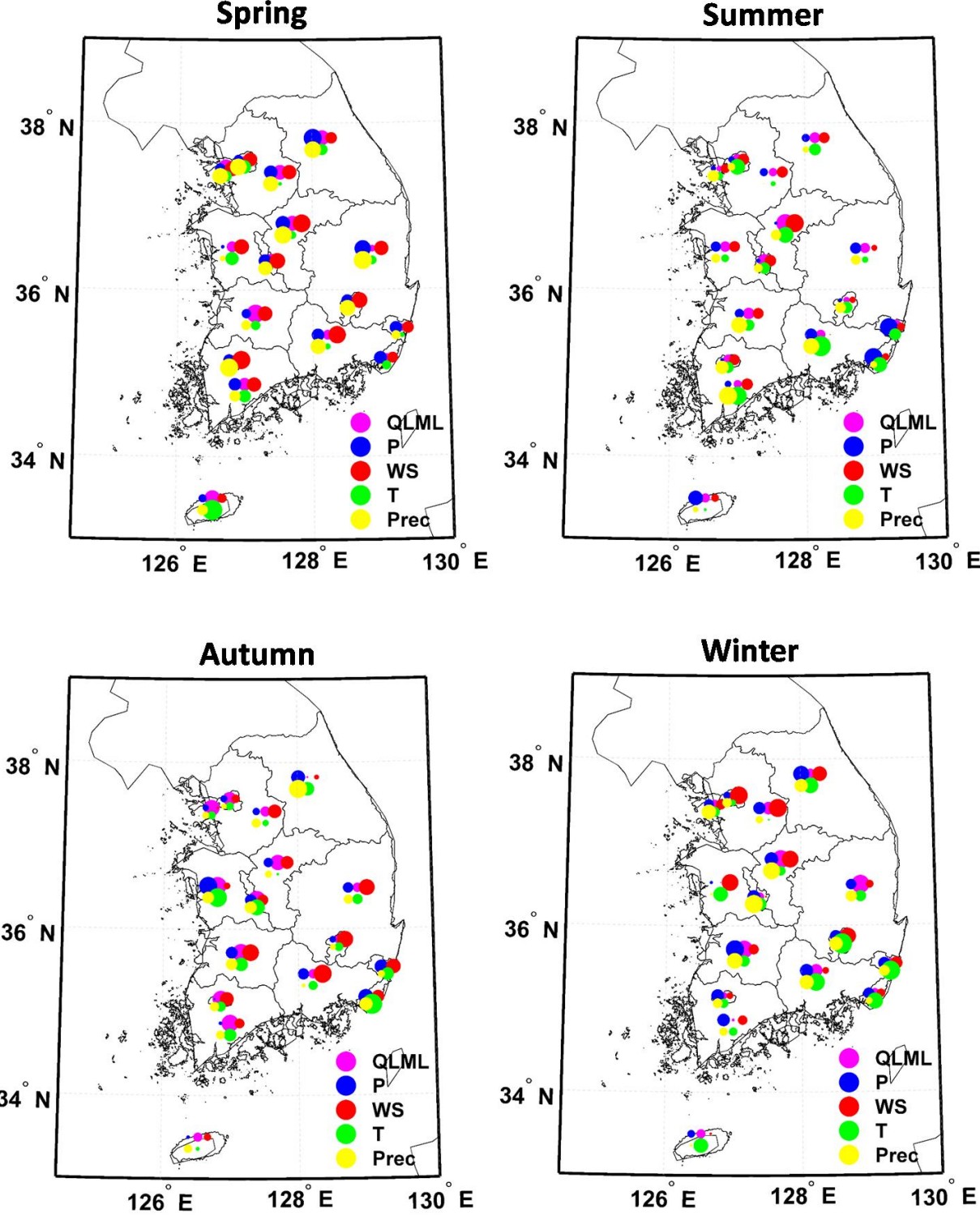

**Figure 13.** MLR model results between $PM_{2.5}$ and the meteorological parameters. The colors represent the type of the meteorological parameter, and the size indicates the magnitude of the influencing parameter. $R^2$ represents the fitting effect of the model, the F-test determines the significance of the model, and the *t*-test determines the significance of the independent variables.

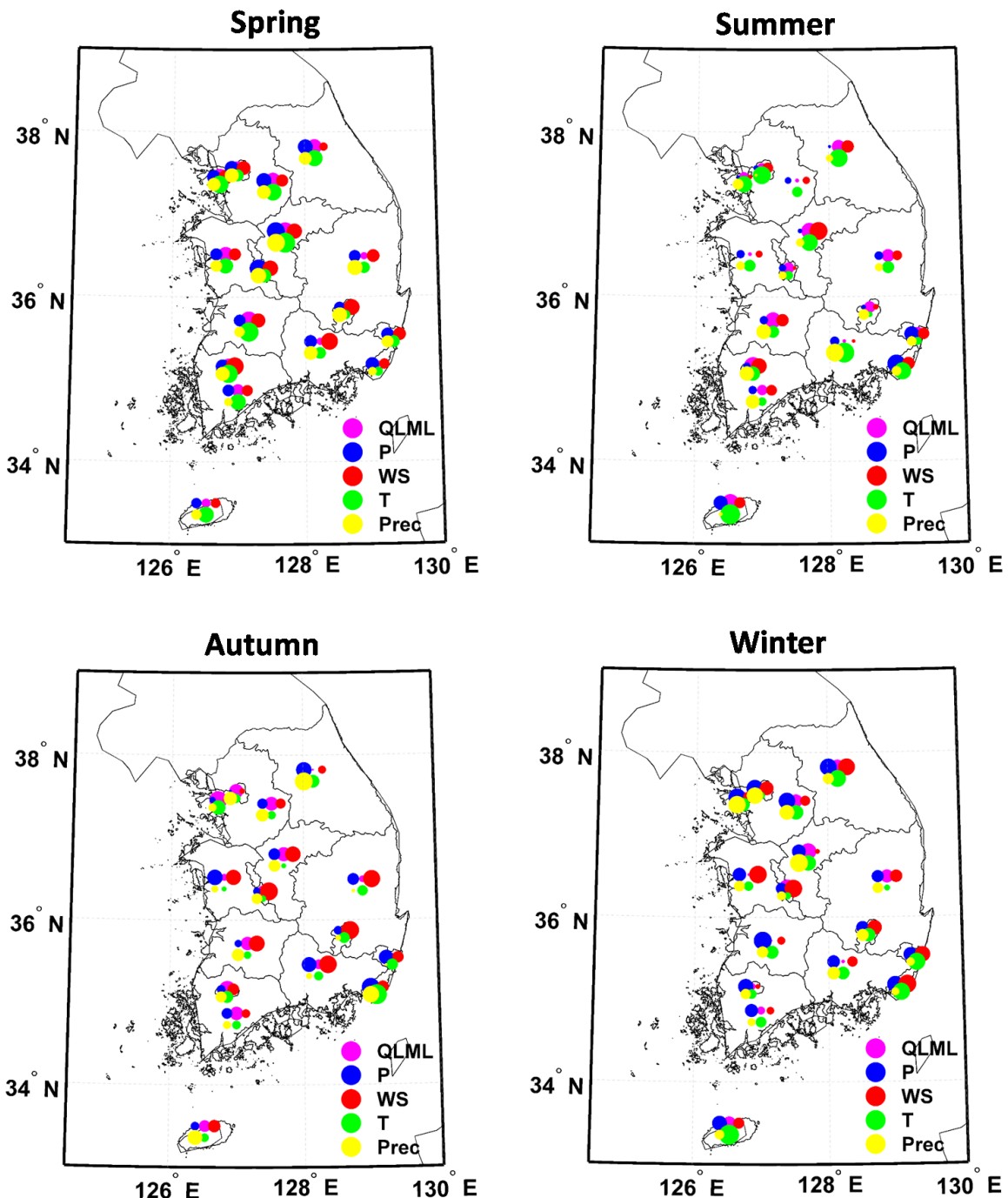

**Figure 14.** MLR model results between $PM_{10}$ and the meteorological parameters. The colors represent the type of the meteorological parameter, and the size indicates the magnitude of the influencing parameter. $R^2$ represents the fitting effect of the model, the F-test determines the significance of the model, and the *t*-test determines the significance of the independent variables.

## 4. Conclusions

We analyzed the spatial and temporal variability in $PM_{2.5}$ and $PM_{10}$ over South Korea during 2015–2020, as well as the relationship between PM, other pollutants, and meteorological parameters. The highest nationally averaged $PM_{2.5}$ concentrations were observed during December and January, and the highest $PM_{10}$ concentrations were observed during March and April. $PM_{2.5}$ and $PM_{10}$ were the lowest during July and August. Seasonal patterns revealed that $PM_{2.5}$ levels were highest in winter, then in spring, autumn, and summer. CO, BC, and $SO_2$ concentrations peaked as well during the winter. In contrast, $PM_{10}$,

$SO_4$, and $O_3$ were highest in the spring and lowest in the summer. High PM concentrations were attributed to emissions from traffic, household heating and cooking, and industries during the winter and spring. During spring and winter, the prevailing westerly and northwesterly winds were relatively strong and transported anthropogenic emissions and dust to Korea from China and Mongolia. The greater Seoul region is characterized by high traffic, population density, and various industrial facilities; therefore, high $PM_{10}$ and $PM_{2.5}$ concentrations were recorded in the region. CO, BC, $SO_2$, and $SO_4$ were also high over the greater Seoul region. The highest $PM_{2.5}$ concentrations were observed over Chungcheongbuk and Jeollabuk due to emissions from biological, semiconductor, and light industries in these provinces. Furthermore, prevailing winds carried emissions from greater Seoul to Chungcheongbuk and Jeollabuk. Western South Korea had higher PM and pollutant concentrations than the eastern regions due to its proximity to China. Due to the occurrence of two heavy haze events in 2019, pollution was high, which was aided by high AOD values. The back trajectories of the Hysplit model described how emissions from Eastern China sources significantly influence pollution in Korea during haze events. The Jeju site was less influenced by foreign pollution. We explored the relationships between $PM_{2.5}$ and $PM_{10}$, meteorological parameters, and other pollutants. $PM_{2.5}$ and $PM_{10}$ concentrations showed negative correlations with temperature, QLML, and precipitation in all seasons. During summer, $PM_{2.5}$ and $PM_{10}$ had strong negative correlations with temperature, QLML, and precipitation. Temperature and QLML tend to be high during summer, and these warm and humid conditions were conducive to strong convection and precipitation. Under heavy precipitation, the wet scavenging effect reduced PM concentrations. During the summer and spring, $PM_{2.5}$ and $PM_{10}$ were inversely related to WS. During these seasons, W was high, and U was low, promoting vertical pollutant dispersion. Conversely, horizontal winds predominated during autumn and winter and horizontal dispersion was the primary driver modulating PM concentrations. $PM_{2.5}$ and $PM_{10}$ were positively correlated with pressure during all seasons except winter. PM is transported upwards by low pressure and high W, whereas PM is accumulated by high pressure and low W. PM was positively correlated with ground heating during all seasons, except autumn, which was attributed to varying seasonal soil moisture content. PM concentrations were positively correlated with CO, BC, $SO_2$, and $SO_4$ in all seasons. A GAM analysis was conducted to identify PM concentration dependencies on meteorological parameters and other pollutants. Furthermore, the MLR model demonstrated the impact of meteorological parameters on PM concentrations and other pollutants at each site throughout the seasons. Spearman's rank correlation, GAM, and MLR were used to investigate the relationship between meteorological factors and PM concentrations, as well as the dominant influencing factors and their effect on pollution levels in each region of Korea during different seasons.

The primary focus of this study was the relationship between meteorological factors and PM concentrations during different months and seasons. However, in the future, we will conduct a separate study to examine the contributions of foreign/transboundary and local air pollution in Korea for various meteorological conditions/events using WRF, CMAQ, and other chemical models.

**Supplementary Materials:** The following supporting information can be downloaded at: https://www.mdpi.com/article/10.3390/rs14194849/s1, Figure S1: Diurnal distribution of aerosol optical depth (AOD) during the severe haze events in (a) January and (b) February–March in 2019. The vertical dotted line in (b) indicates the separation between the two months; Figure S2: (a) Surface temperature and relative humidity profiles and (b) surface pressure and wind profiles on 13–14 January 2019; the dashed lines indicate isotherms. (c) Surface temperature and relative humidity profiles and (d) surface pressure and wind profiles on 4–5 March 2019. The dashed lines indicate isotherms, the white spaces represent the absence of data, and the solid-colored lines represent isobars; Figure S3: Diurnal distribution of the particulate matter (PM) concentration from January to March 2020. (a) $PM_{2.5}$ concentrations and (b) $PM_{10}$ concentrations; Figure S4: Trajectory frequencies (percentage of airmass transport) over the four sites during 20 February–10 March 2020; Figure S5: Trajectory frequencies (percentage of airmass transport) over the four sites during 1–15 January 2020;

Figure S6: Spatial distribution of seasonal mean wind speed (m s$^{-1}$) and direction over East Asia (2015–2020). (a) spring, (b) summer, (c) autumn, and (d) winter; Figure S7: Mean annual distribution of the meteorological parameters impacting air pollution in South Korea (2015–2020). The straight red line indicates the linear fit of the values. (a) temperature, (b) specific humidity, (c) precipitation, (d) wind speed, (e) pressure, (f) ground heating, (g) horizontal (zonal) wind vector, (h) vertical wind vector, (i) sensible heat flux, and (j) latent heat flux; Figure S8: MLR model results between CO and the meteorological parameters. The colors represent the type of the meteorological parameter, and the size indicates the magnitude of the influencing parameter. $R^2$ represents the fitting effect of the model, the F-test determines the significance of the model, and the *t*-test determines the significance of the independent variables; Figure S9: MLR model results between BC and the meteorological parameters. The colors represent the type of the meteorological parameter, and the size indicates the magnitude of the influencing parameter. $R^2$ represents the fitting effect of the model, the F-test determines the significance of the model, and the *t*-test determines the significance of the independent variables; Figure S10: MLR model results between $SO_4$ and the meteorological parameters. The colors represent the type of the meteorological parameter, and the size indicates the magnitude of the influencing parameter. $R^2$ represents the fitting effect of the model, the F-test determines the significance of the model, and the *t*-test determines the significance of the independent variables; Figure S11: MLR model results between $SO_2$ and the meteorological parameters. The colors represent the type of the meteorological parameter and size indicates the magnitude of the influencing parameter. $R^2$ represents the fitting effect of the model, the F-test determines the significance of the model, and the *t*-test determines the significance of the independent variables; Figure S12: MLR model results between $O_3$ and the meteorological parameters. The colors represent the type of the meteorological parameter, and the size indicates the magnitude of the influencing parameter. $R^2$ represents the fitting effect of the model, the F-test determines the significance of the model, and the *t*-test determines the significance of the independent variables.

**Author Contributions:** S.A. performed data analyses and conceptualization and wrote the first manuscript draft. S.L. provided helpful discussions on the analyses of data, conceptualization, methodology, and review and edited the manuscript. K.-S.C. and T.J.Y. reviewed and edited the manuscript. All authors have read and agreed to the published version of the manuscript.

**Funding:** Research for this paper was carried out under the KICT Research Program (20220136-001, Development of 3D Fine Dust Estimating Technology based on AI Image Analysis) funded by the Ministry of Science and ICT.

**Data Availability Statement:** The PM$_{2.5}$ data used in the present study are available at Air Korea, https://www.airkorea.or.kr/web/pmRelay?itemCode=10007&pMENU_NO=108 (accessed on 8 July 2021). The data used for meteorological parameters and other pollutants is available at MERRA-2, https://disc.sci.gsfc.nasa.gov/datasets?keywords=%22MERRA-%22&page=1&source=Models%2FAnalyses%20MERRA-2 (accessed on 10 July 2021).

**Acknowledgments:** The authors would like to thank GMAO NASA for providing the MERRA-2 surface datasets used in this study.

**Conflicts of Interest:** The authors declare no conflict of interest.

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
