# Peer review of "Particulate Matter Concentrations over South Korea: Impact of Meteorology and Other Pollutants"

_remotesensing, doi:10.3390/rs14194849_

Round 1

Reviewer 1 Report (Previous Reviewer 1)

My question has basically been solved.

Author Response

We would like to thank the Reviewer for his/her valuable contribution, for the many useful suggestions and corrections and for his/her constructive comments that lead, in our opinion, to the improvement of the manuscript.

We modified/revised the manuscript according to the points raised by the Reviewer as follows.

i) We made English corrections throughout the manuscript.

ii) We revised the introduction and included relevant background research.

iii) We revised and added references relevant to the current study.

iv). The research design is properly described.

v). The methods used in the manuscript are adequately described.

vi). The results have been modified and clearly presented.

vii). Based on the results, we revised and clearly presented the conclusions.

Reviewer 2 Report (Previous Reviewer 3)

I do not have other recommendations for your paper.

Author Response

We would like to thank the Reviewer for his/her valuable contribution.

We thank the Reviewer for accepting the paper.

This manuscript is a resubmission of an earlier submission. The following is a list of the peer review reports and author responses from that submission.

Round 1

Reviewer 1 Report

In this paper, the author has compared PM concentrations with meteorological conditions and the concentrations of other airborne pollutants over South Korea from 2015 to 2020, using different linear and non-linear models such as linear regression, generalized additive, and multivariable linear regression models.

I think a very important question is that, the authors contributed their research article to remote sensing, which is biased towards remote sensing methods and remote sensing data applications. But the data in this article is PM data collected by the ground station, namely KMOE monitoring network, Besides, the reanalysis and meteorological data they used is also not the traditional remote sensing data. At the same time, the authors mixed too much political content. They pointed out in the introduction and results part that China, Mongolia, and Russia are the main pollution sources. In general, South Korea is in the downwind location of China, Mongolia, and Russia under certain seasonal or meteorological conditions. As a matter of fact, due to the influence of wind, there is indeed pollution from their mentioned aera in China, Mongolia, and Russia, but this kind of external pollution only accounts for a part or a small part of the annual pollution. Local pollution can also occupy an important position.

I also read the reference 3, which studied the problem of pm concentration rising in Seoul in 2017 during a certain time. Thus, the authors need to verify their own view with Long-time series data, rather than simple quotation or a short-time analysis.

The following comments for reference only

1.     Please explain the PM concentration data obtained by the Korean Ministry of Environment (KMOE) monitoring network used in this article and the quality and accuracy of the data.

2.     Regarding the distribution of PM2.5 in the administrative region of South Korea, line 298 wrote that these were attributed to the severe dust event that occurred during the year. Why the event affected the distribution? And how it affected the distribution?

3.     I think the color and format of the figures in your article needs to be improved

Reviewer 2 Report

The relevance of the manuscript is the lack of data and reported analysis for South Korea. But some aspects can be improved:

- It is missing, in the introduction,  other relevant air quality studies for the region;

- in the description of the region, the emission inventory should be included to quantitatively indicate the importance of each source for air quality.

- In the methods, the inclusion of BC as an independent variable in the regression of PM should be discussed. BC usually correlates well with PM and this can be tested and eventually not considered in the regression.

- Why do the authors use the GAM and then the MLR models?

- The authors said that "Organic carbon, elemental carbon,  and PM are transported to Korea by turbulent flow". How could they verify this? What induced the turbulent flow?

- In Figure 8, the linear fit is not adequate for all the pollutants, it has no meaning for ozone and SO2.  And the linear fit should include the confidence curve.

- In Figures 14 and 15 it is no possible to have the magnitude of the influencing  parameter only by observing the figure.

Reviewer 3 Report

I found a good mathematical structure and good explanation even if the models used in your analysis are used by more time. There are many papers that have used the same models. Anyway, it could be used by political makers to protect the population. I suggest to you two papers to add to strengthen some concepts:

"Analysis of vertical profile of particulates dispersion in function of the aerodynamic diameter at a congested road in Catania";

 "Experimental analysis of a plume dispersion around obstacles."

Furthermore, you must sort out the formula in row 132, but more importantly reduced the number of graphs. It is difficult to follow all the graphs, I suggest uniting fig. 9 and fig.10. The PM10 and PM2.5 values can be represented in only graph, obviously for each monitoring station. If you improve these weak points, In my opinion, this work has a good potential to be published in this journal.